# SOBOLEV GAN

**Youssef Mroueh[†], Chun-Liang Li[∘,⋆], Tom Sercu[†,⋆], Anant Raj[◇,⋆] & Yu Cheng[†]**
† IBM Research AI
∘ Carnegie Mellon University
◇ Max Planck Institute for Intelligent Systems
⋆ denotes Equal Contribution
{mroueh,chengyu}@us.ibm.com, chunlial@cs.cmu.edu,
tom.sercu1@ibm.com,anant.raj@tuebingen.mpg.de

## ABSTRACT

We propose a new Integral Probability Metric (IPM) between distributions: the Sobolev IPM. The Sobolev IPM compares the mean discrepancy of two distributions for functions (critic) restricted to a Sobolev ball defined with respect to a dominant measure $\mu$. We show that the Sobolev IPM compares two distributions in high dimensions based on weighted conditional Cumulative Distribution Functions (CDF) of each coordinate on a leave one out basis. The Dominant measure $\mu$ plays a crucial role as it defines the support on which conditional CDFs are compared. Sobolev IPM can be seen as an extension of the one dimensional Von-Mises Cramér statistics to high dimensional distributions. We show how Sobolev IPM can be used to train Generative Adversarial Networks (GANs). We then exploit the intrinsic conditioning implied by Sobolev IPM in text generation. Finally we show that a variant of Sobolev GAN achieves competitive results in semi-supervised learning on CIFAR-10, thanks to the smoothness enforced on the critic by Sobolev GAN which relates to Laplacian regularization. [1]

## 1 INTRODUCTION

In order to learn Generative Adversarial Networks (Goodfellow et al., 2014), it is now well established that the generator should mimic the distribution of real data, in the sense of a certain discrepancy measure. Discrepancies between distributions that measure the goodness of the fit of the neural generator to the real data distribution has been the subject of many recent studies (Arjovsky & Bottou, 2017; Nowozin et al., 2016; Kaae Sønderby et al., 2017; Mao et al., 2017; Arjovsky et al., 2017; Gulrajani et al., 2017; Mroueh et al., 2017; Mroueh & Sercu, 2017; Li et al., 2017), most of which focus on training stability.

In terms of data modalities, most success was booked in plausible natural image generation after the introduction of Deep Convolutional Generative Adversarial Networks (DCGAN) (Radford et al., 2015). This success is not only due to advances in training generative adversarial networks in terms of loss functions (Arjovsky et al., 2017) and stable algorithms, but also to the representation power of convolutional neural networks in modeling images and in finding sufficient statistics that capture the *continuous* density function of natural images. When moving to neural generators of *discrete sequences* generative adversarial networks theory and practice are still not very well understood. Maximum likelihood pre-training or augmentation, in conjunction with the use of reinforcement learning techniques were proposed in many recent works for training GAN for discrete sequences generation (Yu et al., 2016; Che et al., 2017; Hjelm et al., 2017; Rajeswar et al., 2017). Other methods included using the Gumbel Softmax trick (Kusner & Hernández-Lobato, 2016) and the use of auto-encoders to generate adversarially discrete sequences from a continuous space (Zhao et al., 2017). *End to end* training of GANs for discrete sequence generation is still an open problem (Press et al., 2017). Empirical successes of end to end training have been reported within the framework of WGAN-GP (Gulrajani et al., 2017), using a proxy for the Wasserstein distance via a

---

[1] Code for semi-supervised learning experiments is available on https://github.com/tomsercu/SobolevGAN-SSL

*pointwise gradient penalty* on the critic. Inspired by this success, we propose in this paper a new Integral Probability Metric (IPM) between distributions that we coin *Sobolev IPM*. Intuitively an IPM (Müller, 1997) between two probability distributions looks for a witness function $f$, called critic, that maximally discriminates between samples coming from the two distributions:

$$\sup_{f \in \mathscr{F}} \mathbb{E}_{x \sim \mathbb{P}} f(x) - \mathbb{E}_{x \sim \mathbb{Q}} f(x).$$

Traditionally, the function $f$ is defined over a function class $\mathscr{F}$ that is independent to the distributions at hand (Sriperumbudur et al., 2012). The Wasserstein-1 distance corresponds for instance to an IPM where the witness functions are defined over the space of Lipschitz functions; The MMD distance (Gretton et al., 2012) corresponds to witness functions defined over a ball in a Reproducing Kernel Hilbert Space (RKHS).

We will revisit in this paper Fisher IPM defined in (Mroueh & Sercu, 2017), which extends the IPM definition to function classes defined with norms that depend on the distributions. Fisher IPM can be seen as restricting the *critic* to a Lebsegue ball defined with respect to a dominant measure $\mu$. The Lebsegue norm is defined as follows:

$$\int_{\mathcal{X}} f^2(x)\mu(x)dx.$$

where $\mu$ is a dominant measure of $\mathbb{P}$ and $\mathbb{Q}$.

In this paper we extend the IPM framework to critics bounded in the Sobolev norm:

$$\int_{\mathcal{X}} \|\nabla_x f(x)\|_2^2 \, \mu(x)dx,$$

In contrast to Fisher IPM, which compares joint *probability density functions* of all coordinates between two distributions, we will show that Sobolev IPM compares *weighted (coordinate-wise) conditional Cumulative Distribution Functions* for all coordinates on a leave on out basis. Matching conditional dependencies between coordinates is crucial for **sequence modeling**.

Our analysis and empirical verification show that the modeling of the conditional dependencies can be built in to the metric used to learn GANs as in Sobolev IPM. For instance, this gives an advantage to Sobolev IPM in comparing sequences over Fisher IPM. Nevertheless, in sequence modeling when we parametrize the critic and the generator with a neural network, we find an interesting tradeoff between the metric used and the architectures used to parametrize the critic and the generator as well as the conditioning used in the generator. The burden of modeling the conditional long term dependencies can be handled by the IPM loss function as in Sobolev IPM (more accurately the choice of the data dependent function class of the critic) or by a simpler metric such as Fisher IPM together with a powerful architecture for the critic that models conditional long term dependencies such as LSTM or GRUs in conjunction with a curriculum conditioning of the generator as done in (Press et al., 2017). Highlighting those interesting tradeoffs between metrics, data dependent functions classes for the critic (Fisher or Sobolev) and architectures is crucial to advance sequence modeling and more broadly structured data generation using GANs.

On the other hand, Sobolev norms have been widely used in manifold regularization in the so called Laplacian framework for semi-supervised learning (SSL) (Belkin et al., 2006). GANs have shown success in semi-supervised learning (Salimans et al., 2016; Dumoulin et al., 2017; Dai et al., 2017; Kumar et al., 2017). Nevertheless, many normalizations and additional tricks were needed. We show in this paper that a variant of Sobolev GAN achieves strong results in semi-supervised learning on CIFAR-10, without the need of any activation normalization in the critic.

The main contributions of this paper can be summarized as follows:

1. We overview in Section 2 different metrics between distribution used in the GAN literature. We then generalize Fisher IPM in Section 3 with a general dominant measure $\mu$ and show how it compares distributions based on their PDFs.

2. We introduce Sobolev IPM in Section 4 by restricting the critic of an IPM to a Sobolev ball defined with respect to a dominant measure $\mu$. We then show that Sobolev IPM defines a discrepancy between weighted (coordinate-wise) conditional CDFs of distributions.

3. The intrinsic conditioning and the CDF matching make Sobolev IPM suitable for discrete sequence matching and explain the success of the gradient pernalty in WGAN-GP and Sobolev GAN in discrete sequence generation.

4. We give in Section 5 an ALM (Augmented Lagrangian Multiplier) algorithm for training Sobolev GAN. Similar to Fisher GAN, this algorithm is stable and does not compromise the capacity of the critic.

5. We show in Appendix A that the critic of Sobolev IPM satisfies an elliptic Partial Differential Equation (PDE). We relate this diffusion to the Fokker-Planck equation and show the behavior of the gradient of the optimal Sobolev critic as a transportation plan between distributions.

6. We empirically study Sobolev GAN in character level text generation (Section 6.1). We validate that the conditioning implied by Sobolev GAN is crucial for the success and stability of GAN in text generation. As a take home message from this study, we see that text generation succeeds either by *implicit conditioning* i.e using Sobolev GAN (or WGAN-GP) together with convolutional critics and generators, or by *explicit conditioning* i.e using Fisher IPM together with recurrent critic and generator and curriculum learning.

7. We finally show in Section 6.2 that a variant of Sobolev GAN achieves competitive semi-supervised learning results on CIFAR-10, thanks to the smoothness implied by the Sobolev regularizer.

## 2 OVERVIEW OF METRICS BETWEEN DISTRIBUTIONS

In this Section, we review different representations of probability distributions and metrics for comparing distributions that use those representations. Those metrics are at the core of training GAN. In what follows, we consider probability measures with a positive weakly differentiable probability density functions (PDF). Let $P$ and $Q$ be two probability measures with PDFs $\mathbb{P}(x)$ and $\mathbb{Q}(x)$ defined on $\mathcal{X} \subset \mathbb{R}^d$. Let $F_{\mathbb{P}}$ and $F_{\mathbb{Q}}$ be the Cumulative Distribution Functions (CDF) of $\mathbb{P}$ and $\mathbb{Q}$ respectively. For $x = (x_1, \ldots, x_d)$, we have:

$$F_{\mathbb{P}}(x) = \int_{-\infty}^{x_1} \ldots \int_{-\infty}^{x_d} \mathbb{P}(u_1, \ldots u_d) du_1 \ldots du_d.$$

The score function of a density function is defined as: $s_{\mathbb{P}}(x) = \nabla_x \log(\mathbb{P}(x)) \in \mathbb{R}^d$.

In this work, we are interested in metrics between distributions that have a variational form and can be written as a suprema of mean discrepancies of functions defined on a specific function class. This type of metrics include $\varphi$-divergences as well as Integral Probability Metrics (Sriperumbudur et al., 2009) and have the following form:

$$d_{\mathscr{F}}(\mathbb{P}, \mathbb{Q}) = \sup_{f \in \mathscr{F}} |\Delta(f; \mathbb{P}, \mathbb{Q})|,$$

where $\mathscr{F}$ is a function class defined on $\mathcal{X}$ and $\Delta$ is a mean discrepancy, $\Delta : \mathscr{F} \to \mathbb{R}$. The variational form given above leads in certain cases to closed form expressions in terms of the PDFs $\mathbb{P}, \mathbb{Q}$ or in terms of the CDFs $F_{\mathbb{P}}, F_{\mathbb{Q}}$ or the score functions $s_{\mathbb{P}}, s_{\mathbb{Q}}$.

In Table 1, we give a comparison of different discrepancies $\Delta$ and function spaces $\mathscr{F}$ used in the literature for GAN training together with our proposed Sobolev IPM. We see from Table 1 that Sobolev IPM, compared to Wasserstein Distance, imposes a tractable smoothness constraint on the critic on points sampled from a distribution $\mu$, rather then imposing a Lipschitz constraint on all points in the space $\mathcal{X}$. We also see that Sobolev IPM is the natural generalization of the Cramér Von-Mises Distance from one dimension to high dimensions. We note that the Energy Distance, a form of Maximum Mean Discrepancy for a special kernel, was used in (Bellemare et al., 2017b) as a generalization of the Cramér distance in GAN training but still needed a gradient penalty in its algorithmic counterpart leading to a mis-specified distance between distributions. Finally it is worth noting that when comparing Fisher IPM and Sobolev IPM we see that while Fisher IPM compares joint PDF of the distributions, Sobolev IPM compares weighted (coordinate-wise) conditional CDFs. As we will see later, this conditioning nature of the metric makes Sobolev IPM suitable for comparing sequences. Note that the Stein metric (Liu et al., 2016; Liu, 2017) uses the score function to match distributions. We will show later how Sobolev IPM relates to the Stein discrepancy (Appendix A).

| | $\Delta(f; \mathbb{P}, \mathbb{Q})$ | $\mathcal{F}$ Function class | $d_{\mathscr{F}}(\mathbb{P}, \mathbb{Q})$ Closed Form |
|---|---|---|---|
| $\varphi$-Divergence (Goodfellow et al., 2014) (Nowozin et al., 2016) | $\mathbb{E}_{x\sim\mathbb{P}} f(x) - \mathbb{E}_{x\sim\mathbb{Q}} \varphi^*(f(x))$ 

 $\varphi^*$ Fenchel Conjugate | $\left\{ f : \mathcal{X} \to \mathbb{R}, f \in \mathrm{dom}_{\varphi^*} \right\}$ | $\mathbb{E}_{x\sim\mathbb{Q}} \left[ \varphi(\frac{\mathbb{P}(x)}{\mathbb{Q}(x)}) \right]$ |
| Wasserstein -1 (Arjovsky et al., 2017) (Gulrajani et al., 2017) | $\mathbb{E}_{x\sim\mathbb{P}} f(x) - \mathbb{E}_{x\sim\mathbb{Q}} f(x)$ | $\left\{ f : \mathcal{X} \to \mathbb{R}, \|f\|_{\mathrm{lip}} \le 1 \right\}$ | $\inf_{\pi\in\Pi(\mathbb{P},\mathbb{Q})} \int_{\mathcal{X}} \|x-y\|_1 \, d\pi(x,y)$ 
 Sinkhorn Divergence 
 (Genevay et al., 2017) |
| MMD (Li et al., 2017) (Li et al., 2015) (Dziugaite et al., 2015) | $\mathbb{E}_{x\sim\mathbb{P}} f(x) - \mathbb{E}_{x\sim\mathbb{Q}} f(x)$ | $\left\{ f : \mathcal{X} \to \mathbb{R}, \|f\|_{\mathscr{H}_k} \le 1 \right\}$ | $\left\| \mathbb{E}_{x\sim\mathbb{P}} k_x - \mathbb{E}_{x\sim\mathbb{Q}} k_x \right\|_{\mathscr{H}_k}$ |
| Stein Discrepancy (Wang & Liu, 2016) | $\mathbb{E}_{x\sim\mathbb{Q}} \left[ T(\mathbb{P}) f(x) \right]$ 
 $T(\mathbb{P}) = (\nabla_x \log(\mathbb{P}(x))^\top + \nabla_x.$ | $\left\{ f : \mathcal{X} \to \mathbb{R}^d \right.$ 
 $f$ smooth with zero 
 boundary condition $\left. \right\}$ | NA in general 
 has a closed form 
 in RKHS |
| Cramér for $d = 1$ (Bellemare et al., 2017a) | $\mathbb{E}_{x\sim\mathbb{P}} f(x) - \mathbb{E}_{x\sim\mathbb{Q}} f(x)$ | $\left\{ f : \mathcal{X} \to \mathbb{R}, \mathbb{E}_{x\sim\mathbb{P}}(\frac{df(x)}{dx})^2 \le 1, \right.$ 
 $f$ smooth with zero 
 boundary condition $\left. \right\}$ | $\sqrt{ \mathbb{E}_{x\sim\mathbb{P}} \left( \frac{F_{\mathbb{P}}(x) - F_{\mathbb{Q}}(x)}{\mathbb{P}(x)} \right)^2 }$ 
 $x \in \mathbb{R}$ |
| $\mu$-Fisher IPM (Mroueh & Sercu, 2017) | $\mathbb{E}_{x\sim\mathbb{P}} f(x) - \mathbb{E}_{x\sim\mathbb{Q}} f(x)$ | $\left\{ f : \mathcal{X} \to \mathbb{R}, f \in \mathscr{L}_2(\mathcal{X},\mu), \right.$ 
 $\mathbb{E}_{x\sim\mu} f^2(x) \le 1 \left. \right\}$ | $\sqrt{ \mathbb{E}_{x\sim\mu} \left( \frac{\mathbb{P}(x) - \mathbb{Q}(x)}{\mu(x)} \right)^2 }$ |
| $\mu$-Sobolev IPM (This work) | $\mathbb{E}_{x\sim\mathbb{P}} f(x) - \mathbb{E}_{x\sim\mathbb{Q}} f(x)$ | $\left\{ f : \mathcal{X} \to \mathbb{R}, f \in W_0^{1,2}(\mathcal{X},\mu), \right.$ 
 $\mathbb{E}_{x\sim\mu} \|\nabla_x f(x)\|^2 \le 1,$ 
 with zero boundary condition $\left. \right\}$ | $\frac{1}{d} \sqrt{ \mathbb{E}_{x\sim\mu} \sum_{i=1}^d \left( \frac{\phi_i(\mathbb{P}) - \phi_i(\mathbb{Q})}{\mu(x)} \right)^2 }$ 
 where $\phi_i(\mathbb{P}) =$ 
 $\mathbb{P}_{X^{-i}}(x^{-i}) F_{\mathbb{P}_{[X_i|X^{-i}=x^{-i}]}}(x_i)$ 
 $x^{-i} = (x_1, \ldots x_{i-1}, x_{i+1}, \ldots x_d)$ |

Table 1: Comparison of different metrics between distributions used for GAN training. References are for papers using those metrics for GAN training.

## 3 GENERALIZING FISHER IPM: PDF COMPARISON

Imposing data-independent constraints on the function class in the IPM framework, such as the Lipschitz constraint in the Wasserstein distance is computationally challenging and intractable for the general case. In this Section, we generalize the Fisher IPM introduced in (Mroueh & Sercu, 2017), where the function class is relaxed to a *tractable data dependent constraint* on the second order moment of the critic, in other words the critic is constrained to be in a Lebsegue ball.

**Fisher IPM.** Let $\mathcal{X} \subset \mathbb{R}^d$ and $\mathscr{P}(\mathcal{X})$ be the space of distributions defined on $\mathcal{X}$. Let $\mathbb{P}, \mathbb{Q} \in \mathscr{P}(\mathcal{X})$, and $\mu$ be a dominant measure of $\mathbb{P}$ and $\mathbb{Q}$, in the sense that

$$\mu(x) = 0 \implies \mathbb{P}(x) = 0 \text{ and } \mathbb{Q}(x) = 0.$$

We assume $\mu$ to be also a distribution in $\mathscr{P}(\mathcal{X})$, and assume $\boldsymbol{\mu(x) > 0}$, $\forall x \in \mathcal{X}$. Let $\mathscr{L}_2(\mathcal{X}, \mu)$ be the space of $\mu$-measurable functions. For $f, g \in \mathscr{L}_2(\mathcal{X}, \mu)$, we define the following dot product and its corresponding norm:

$$\langle f, g \rangle_{\mathscr{L}_2(\mathcal{X},\mu)} = \int_{\mathcal{X}} f(x) g(x) \mu(x) dx, \quad \|f\|_{\mathscr{L}_2(\mathcal{X},\mu)} = \sqrt{\int_{\mathcal{X}} f^2(x) \mu(x) dx}.$$

Note that $\mathscr{L}_2(\mathcal{X}, \mu)$, can be formally defined as follows:

$$\mathscr{L}_2(\mathcal{X}, \mu) = \{ f : \mathcal{X} \to \mathbb{R} \text{ s.t } \|f\|_{\mathscr{L}_2(\mathcal{X},\mu)} < \infty \}.$$

We define the unit Lebesgue ball as follows:

$$\mathbb{B}_2(\mathcal{X}, \mu) = \{f \in \mathscr{L}_2(\mathcal{X}, \mu), \|f\|_{\mathscr{L}_2(\mathcal{X},\mu)} \leq 1\}.$$

Fisher IPM defined in (Mroueh & Sercu, 2017), searches for the critic function *in the Lebesgue Ball* $\mathbb{B}_2(\mathcal{X}, \mu)$ that maximizes the mean discrepancy between $\mathbb{P}$ and $\mathbb{Q}$. Fisher GAN (Mroueh & Sercu, 2017) was originally formulated specifically for $\mu = \frac{1}{2}(\mathbb{P} + \mathbb{Q})$. We consider here a general $\mu$ as long as it dominates $\mathbb{P}$ and $\mathbb{Q}$. We define Generalized Fisher IPM as follows:

$$\mathscr{F}_\mu(\mathbb{P}, \mathbb{Q}) = \sup_{f \in \mathbb{B}_2(\mathcal{X},\mu)} \mathbb{E}_{x \sim \mathbb{P}} f(x) - \mathbb{E}_{x \sim \mathbb{Q}} f(x) \tag{1}$$

Note that:

$$\mathbb{E}_{x \sim \mathbb{P}} f(x) - \mathbb{E}_{x \sim \mathbb{Q}} f(x) = \left\langle f, \frac{\mathbb{P} - \mathbb{Q}}{\mu} \right\rangle_{\mathscr{L}_2(\mathcal{X},\mu)}.$$

Hence Fisher IPM can be written as follows:

$$\mathscr{F}_\mu(\mathbb{P}, \mathbb{Q}) = \sup_{f \in \mathbb{B}_2(\mathcal{X},\mu)} \left\langle f, \frac{\mathbb{P} - \mathbb{Q}}{\mu} \right\rangle_{\mathscr{L}_2(\mathcal{X},\mu)} \tag{2}$$

We have the following result:

**Theorem 1** (Generalized Fisher IPM). *The Fisher distance and the optimal critic are as follows:*

*1. The Fisher distance is given by:*

$$\mathscr{F}_\mu(\mathbb{P}, \mathbb{Q}) = \left\| \frac{\mathbb{P} - \mathbb{Q}}{\mu} \right\|_{\mathscr{L}_2(\mathcal{X},\mu)} = \sqrt{\mathbb{E}_{x \sim \mu} \left( \frac{\mathbb{P}(x) - \mathbb{Q}(x)}{\mu(x)} \right)^2}.$$

*2. The optimal $f_\chi$ achieving the Fisher distance $\mathscr{F}_\mu(\mathbb{P}, \mathbb{Q})$ is:*

$$f_\chi = \frac{1}{\mathscr{F}(\mathbb{P}, \mathbb{Q})} \frac{\mathbb{P} - \mathbb{Q}}{\mu}, \mu \text{ almost surely.}$$

*Proof of Theorem 1.* From Equation (2), the optimal $f_\chi$ belong to the intersection of the hyperplane that has normal $n = \frac{\mathbb{P} - \mathbb{Q}}{\mu}$, and the ball $\mathbb{B}_2(\mathcal{X}, \mu)$, hence $f_\chi = \frac{n}{\|n\|_{\mathscr{L}_2(\mathcal{X},\mu)}}$. Hence $\mathscr{F}(\mathbb{P}, \mathbb{Q}) = \|n\|_{\mathscr{L}_2(\mathcal{X},\mu)}$. $\square$

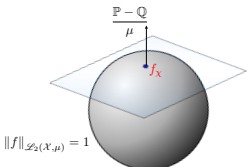

We see from Theorem 1 the role of the dominant measure $\mu$: the optimal critic is defined with respect to this measure and the overall Fisher distance can be seen as an average weighted distance between *probability density functions*, where the average is taken on points sampled from $\mu$. We give here some choices of $\mu$:

1. For $\mu = \frac{1}{2}(\mathbb{P} + \mathbb{Q})$, we obtain the symmetric chi-squared distance as defined in (Mroueh & Sercu, 2017).

2. $\mu_{GP}$, the implicit distribution defined by the interpolation lines between $\mathbb{P}_r$ and $\mathbb{Q}_\theta$ as in (Gulrajani et al., 2017).

3. When $\mu$ does not dominate $\mathbb{P}$, and $\mathbb{Q}$, we obtain a non symmetric divergence. For example for $\mu = \mathbb{P}$, $\mathscr{F}_\mathbb{P}^2(\mathbb{P}, \mathbb{Q}) = \int_\mathcal{X} \frac{(\mathbb{P}(x) - \mathbb{Q}(x))^2}{\mathbb{P}(x)} dx$. We see here that for this particular choice we obtain the Pearson divergence.

## 4  SOBOLEV IPM

In this Section, we introduce the Sobolev IPM. In a nutshell, the Sobolev IPM constrains the critic function to belong to a ball in the restricted Sobolev Space. In other words we constrain the norm of the gradient of the critic $\nabla_x f(x)$. We will show that by moving from a Lebesgue constraint as in Fisher IPM to a Sobolev constraint as in Sobolev IPM, the metric changes from a joint PDF matching to weighted (ccordinate-wise) conditional CDFs matching. The intrinsic conditioning built in to the Sobolev IPM and the comparison of cumulative distributions makes Sobolev IPM suitable for comparing discrete sequences.

### 4.1 DEFINITION AND EXPRESSION OF SOBOLEV IPM IN TERMS OF COORDINATE CONDITIONAL CDFS

We will start by recalling some definitions on Sobolev Spaces. We assume in the following that $\mathcal{X}$ is compact and consider functions in the Sobolev space $W^{1,2}(\mathcal{X}, \mu)$:

$$W^{1,2}(\mathcal{X}, \mu) = \left\{ f : \mathcal{X} \to \mathbb{R}, \int_{\mathcal{X}} \|\nabla_x f(x)\|^2 \mu(x) dx < \infty \right\},$$

We restrict ourselves to functions in $W^{1,2}(\mathcal{X}, \mu)$ vanishing at the boundary, and note this space $W_0^{1,2}(\mathcal{X}, \mu)$. Note that in this case:

$$\|f\|_{W_0^{1,2}(\mathcal{X}, \mu)} = \sqrt{\int_{\mathcal{X}} \|\nabla_x f(x)\|^2 \mu(x) dx}$$

defines a semi-norm. We can similarly define a dot product in $W_0^{1,2}(\mathcal{X}, \mu)$, for $f, g \in W_0^{1,2}(\mathcal{X}, \mu)$:

$$\langle f, g \rangle_{W_0^{1,2}(\mathcal{X}, \mu)} = \int_{\mathcal{X}} \langle \nabla_x f(x), \nabla_x g(x) \rangle_{\mathbb{R}^d} \mu(x) dx.$$

Hence we define the following Sobolev IPM, by restricting the critic of the mean discrepancy to the Sobolev unit ball :

$$\mathcal{S}_{\boldsymbol{\mu}}(\mathbb{P}, \mathbb{Q}) = \sup_{f \in W_0^{1,2}, \|f\|_{W_0^{1,2}(\mathcal{X}, \mu)} \leq 1} \left\{ \mathbb{E}_{x \sim \mathbb{P}} f(x) - \mathbb{E}_{x \sim \mathbb{Q}} f(x) \right\}. \tag{3}$$

When compared to the Wasserstein distance, the Sobolev IPM given in Equation (3) uses a data dependent gradient constraint (depends on $\mu$) rather than a data independent Lipchitz constraint. Let $F_{\mathbb{P}}$ and $F_{\mathbb{Q}}$ be *the cumulative distribution functions* of $\mathbb{P}$ and $\mathbb{Q}$ respectively. We have:

$$\mathbb{P}(x) = \frac{\partial^d}{\partial x_1 \ldots \partial x_d} F_{\mathbb{P}}(x), \tag{4}$$

and we define

$$D^{-i} = \frac{\partial^{d-1}}{\partial x_1 \ldots \partial x_{i-1} \partial x_{i+1} \ldots \partial x_d}, \text{ for } i = 1 \ldots d.$$

$D^{-i}$ computes the $(d-1)$ high-order partial derivative excluding the variable $i$.

Our main result is presented in Theorem 2. Additional theoretical results are given in Appendix A. All proofs are given in Appendix B.

**Theorem 2** (Sobolev IPM). *Assume that $F_{\mathbb{P}}$, and $F_{\mathbb{Q}}$ and its $d$ derivatives exist and are continuous: $F_{\mathbb{P}}$ and $F_{\mathbb{Q}} \in C^d(\mathcal{X})$. Define the differential operator $D^-$ :*

$$D^- = (D^{-1}, \ldots D^{-d}).$$

*For $x = (x_1, \ldots x_{i-1}, x_i, x_{i+1}, \ldots x_d)$, let $x^{-i} = (x_1, \ldots x_{i-1}, x_{i+1}, \ldots x_d)$.*

*The Sobolev IPM given in Equation (3) has the following equivalent forms:*

1. *Sobolev IPM as comparison of high order partial derivatives of CDFs. The Sobolev IPM has the following form:*

$$\mathcal{S}_\mu(\mathbb{P}, \mathbb{Q}) = \frac{1}{d} \sqrt{\int_{\mathcal{X}} \frac{\sum_{i=1}^d (D^{-i} F_{\mathbb{P}}(x) - D^{-i} F_{\mathbb{Q}}(x))^2}{\mu(x)} dx}.$$

2. *Sobolev IPM as comparison of weighted (coordinate-wise) conditional CDFs. The Sobolev IPM can be written in the following equivalent form:*

$$\mathcal{S}_\mu^2(\mathbb{P}, \mathbb{Q}) = \frac{1}{d^2} \mathbb{E}_{x \sim \mu} \sum_{i=1}^d \left( \frac{\mathbb{P}_{X^{-i}}(x^{-i}) F_{\mathbb{P}_{[X_i|X^{-i}=x^{-i}]}}(x_i) - \mathbb{Q}_{X^{-i}}(x^{-i}) F_{\mathbb{Q}_{[X_i|X^{-i}=x^{-i}]}}(x_i)}{\mu(x)} \right)^2.$$

$$\tag{5}$$

3. *The optimal critic $f^*$ satisfies the following identity:*

$$\nabla_x f^*(x) = \frac{1}{d\mathcal{S}_\mu(\mathbb{P}, \mathbb{Q})} \frac{D^- F_{\mathbb{Q}}(x) - D^- F_{\mathbb{P}}(x)}{\mu(x)}, \boldsymbol{\mu} - almost\ surely. \qquad (6)$$

**Sobolev IPM Approximation.** Learning in the whole Sobolev space $W_0^{1,2}$ is challenging hence we need to restrict our function class to a hypothesis class $\mathscr{H}$, such as neural networks. We assume in the following that functions in $\mathscr{H}$ vanish on the boundary of $\mathcal{X}$, and restrict the optimization to the function space $\mathscr{H}$. $\mathscr{H}$ can be a Reproducing Kernel Hilbert Space as in the MMD case or parametrized by a neural network. Define the Sobolev IPM approximation in $\mathscr{H}$:

$$\mathcal{S}_{\mathscr{H},\mu}(\mathbb{P}, \mathbb{Q}) = \sup_{f \in \mathscr{H}, \|f\|_{W_0^{1,2}} \leq 1} \left\{ \mathbb{E}_{x \sim \mathbb{P}} f(x) - \mathbb{E}_{x \sim \mathbb{Q}} f(x) \right\} \qquad (7)$$

The following Lemma shows that the Sobolev IPM approximation in $\mathscr{H}$ is proportional to Sobolev IPM. The tightness of the approximation of the Sobolev IPM is governed by the tightness of the approximation of the optimal Sobolev Critic $f^*$ in $\mathscr{H}$. This approximation is measured in the Sobolev sense, using the Sobolev dot product.

**Lemma 1** (Sobolev IPM Approximation in a Hypothesis Class)**.** *Let $\mathscr{H}$ be a function space with functions vanishing at the boundary. For any $f \in \mathscr{H}$ and for $f^*$ the optimal critic in $W_0^{1,2}$, we have:*

$$\mathcal{S}_{\mathscr{H},\mu}(\mathbb{P}, \mathbb{Q}) = \mathcal{S}_\mu(\mathbb{P}, \mathbb{Q}) \sup_{f \in \mathscr{H}, \|f\|_{W_0^{1,2}(\mathcal{X},\mu)} \leq 1} \int_{\mathcal{X}} \langle \nabla_x f(x), \nabla_x f^*(x) \rangle_{\mathbb{R}^d} \mu(x) dx.$$

Note that this Lemma means that the Sobolev IPM is well approximated if the space $\mathscr{H}$ has an enough representation power to express $\nabla_x f^*(x)$. This is parallel to the Fisher IPM approximation (Mroueh & Sercu, 2017) where it is shown that the Fisher IPM approximation error is proportional to the critic approximation in the Lebesgue sense. Having in mind that the gradient of the critic is the information that is passed on to the generator, we see that this convergence in the Sobolev sense to the optimal critic is an important property for GAN training.

**Relation to Fokker-Planck Diffusion.** We show in Appendix A that the optimal Sobolev critic is the solution of the following elliptic PDE (with zero boundary conditions):

$$\frac{\mathbb{P} - \mathbb{Q}}{\mathcal{S}_\mu(\mathbb{P}, \mathbb{Q})} = -\text{div}(\mu(x)\nabla_x f(x)). \qquad (8)$$

We further link the elliptic PDE given in Equation (8) and the Fokker-Planck diffusion. As we illustrate in Figure 2(b) the gradient of the critic defines a transportation plan for moving the distribution mass from $\mathbb{Q}$ to $\mathbb{P}$.

**Discussion of Theorem 2.** We make the following remarks on Theorem 2:

1. From Theorem 2, we see that the Sobolev IPM compares $d$ higher order partial derivatives of the cumulative distributions $F_{\mathbb{P}}$ and $F_{\mathbb{Q}}$, while Fisher IPM compares the probability density functions.

2. The dominant measure $\mu$ plays a similar role to Fisher:

$$\mathcal{S}_\mu^2(\mathbb{P}, \mathbb{Q}) = \frac{1}{d^2} \sum_{i=1}^d \mathbb{E}_{x \sim \mu} \left( \frac{D^{-i} F_{\mathbb{P}}(x) - D^{-i} F_{\mathbb{Q}}(x)}{\mu(x)} \right)^2,$$

the average distance is defined with respect to points sampled from $\mu$.

3. **Comparison of coordinate-wise Conditional CDFs.** We note in the following $x^{-i} = (x_1, \ldots x_{i-1}, x_{i+1}, \ldots x_d)$. Note that we have:

$$D^{-i}F_{\mathbb{P}}(x) = \frac{\partial^{d-1}}{\partial x_1 \ldots \partial x_{i-1} \partial x_{i+1} \ldots \partial x_d} \int_{-\infty}^{x_1} \ldots \int_{-\infty}^{x_d} \mathbb{P}(u_1 \ldots u_d) du_1 \ldots du_d$$

$$= \int_{-\infty}^{x_i} \mathbb{P}(x_1, \ldots, x_{i-1}, u, x_{i+1}, \ldots, x_d) du$$

$$= \mathbb{P}_{X^{-i}}(x_1, \ldots, x_{i-1}, x_{i+1}, \ldots x_d) \int_{-\infty}^{x_i} \mathbb{P}_{[X_i|X^{-i}=x^{-i}]}(u|x_1, \ldots, x_{i-1}, x_{i+1}, \ldots x_d) du$$

(Using Bayes rule)

$$= \mathbb{P}_{X^{-i}}(x^{-i}) F_{\mathbb{P}_{[X_i|X^{-i}=x^{-i}]}}(x_i),$$

Note that for each $i$, $D^{-i}F_{\mathbb{P}}(x)$ is the cumulative distribution of the variable $X_i$ given the other variables $X^{-i} = x^{-i}$, weighted by the density function of $X^{-i}$ at $x^{-i}$. This leads us to the form given in Equation 5.

We see that the Sobolev IPM compares for each dimension $i$ the conditional cumulative distribution of each variable given the other variables, weighted by their density function. We refer to this as comparison of coordinate-wise CDFs on a leave one out basis. From this we see that we are comparing CDFs, which are better behaved on discrete distributions. Moreover, the conditioning built in to this metric will play a crucial role in comparing sequences as the conditioning is important in this context (See section 6.1).

### 4.2 ILLUSTRATIVE EXAMPLES

**Sobolev IPM / Cramér Distance and Wasserstein-1 in one Dimension.** In one dimension, Sobolev IPM is the Cramér Distance (for $\mu$ uniform on $\mathcal{X}$, we note this $\mu := 1$). While Sobolev IPM in one dimension measures the discrepancy between CDFs, the one dimensional Wasserstein-$p$ distance measures the discrepancy between inverse CDFs:

$$\mathcal{S}_{\mu:=1}^2(\mathbb{P}, \mathbb{Q}) = \int_{\mathcal{X}} (F_{\mathbb{P}}(x) - F_{\mathbb{Q}}(x))^2 dx \text{ versus } W_p^p(\mathbb{P}, \mathbb{Q}) = \int_0^1 |F_{\mathbb{P}}^{-1}(u) - F_{\mathbb{Q}}^{-1}(u)|^p du,$$

Recall also that the Fisher IPM for uniform $\mu$ is given by :

$$\mathscr{F}_{\mu:=1}^2(\mathbb{P}, \mathbb{Q}) = \int_{\mathcal{X}} (\mathbb{P}(x) - \mathbb{Q}(x))^2 dx.$$

Consider for instance two point masses $\mathbb{P} = \delta_{a_1}$ and $\mathbb{Q} = \delta_{a_2}$ with $a_1, a_2 \in \mathbb{R}$. The rationale behind using Wasserstein distance for GAN training is that since it is a weak metric, for far distributions Wasserstein distance provides some signal (Arjovsky et al., 2017). In this case, it is easy to see that $W_1^1(\mathbb{P}, \mathbb{Q}) = \mathcal{S}_{\mu:=1}^2 = |a_1 - a_2|$, while $\mathscr{F}_{\mu:=1}^2(\mathbb{P}, \mathbb{Q}) = 2$. As we see from this simple example, *CDF* comparison is more suitable than PDF for comparing distributions on *discrete spaces*. See Figure 1, for a further discussion of this effect in the GAN context.

**Sobolev IPM between two 2D Gaussians.** We consider $\mathbb{P}$ and $\mathbb{Q}$ to be two dimensional Gaussians with means $\mu_1$ and $\mu_2$ and covariances $\Sigma_1$ and $\Sigma_2$. Let $(x, y)$ be the coordinates in 2D. We note $F_{\mathbb{P}}$ and $F_{\mathbb{Q}}$ the CDFs of $\mathbb{P}$ and $\mathbb{Q}$ respectively. We consider in this example $\mu = \frac{\mathbb{P}+\mathbb{Q}}{2}$. We know from Theorem 2 that the gradient of the Sobolev optimal critic is proportional to the following vector field:

$$\nabla f^*(x, y) \, \alpha \, \frac{1}{\mu(x, y)} \begin{bmatrix} \frac{\partial}{\partial y}(F_{\mathbb{Q}}(x, y) - F_{\mathbb{P}}(x, y)) \\ \frac{\partial}{\partial x}(F_{\mathbb{Q}}(x, y) - F_{\mathbb{P}}(x, y)) \end{bmatrix} \quad (9)$$

In Figure 2 we consider $\mu_1 = [1, 0], \Sigma_1 = \begin{bmatrix} 1.9 & 0.8 \\ 0.8 & 1.3 \end{bmatrix} \mu_2 = [1, -2], \Sigma_2 = \begin{bmatrix} 1.9 & -0.8 \\ -0.8 & 1.3 \end{bmatrix}$.

In Figure 2(a) we plot the numerical solution of the PDE satisfied by the optimal Sobolev critic given in Equation (8), using MATLAB solver for elliptic PDEs (more accurately we solve $-div(\mu(x)\nabla_x f(x)) = \mathbb{P}(x) - \mathbb{Q}(x)$, hence we obtain the solution of Equation (8) up to a normalization constant ($\frac{1}{\mathcal{S}_{\mu}(\mathbb{P}, \mathbb{Q})}$)). We numerically solve the PDE on a rectangle with zero boundary

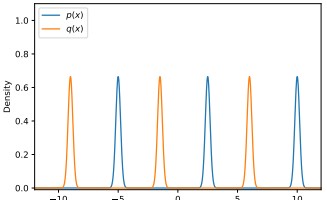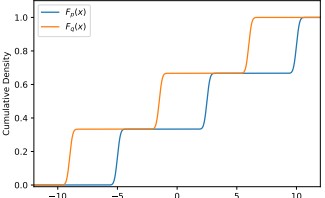

(a) Smoothed discrete densities: PDF versus CDF of smoothed discrete densities with non overlapping supports.

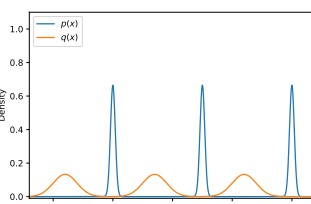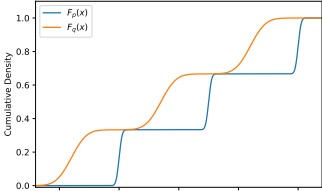

(b) Smoothed Discrete and Continuous densities: PDF versus CDF of a smoothed discrete density and a continuous density with non overlapping supports.

Figure 1: In the GAN context for example in text generation, we have to match a (smoothed) discrete real distribution and a continuous generator. In this case, the CDF matching enabled by Sobolev IPM gives non zero discrepancy between a (smoothed) discrete and a continuous density even if the densities have disjoint supports. This ensures non vanishing gradients of the critic.

conditions. We see that the optimal Sobolev critic separates the two distributions well. In Figure 2(b) we then numerically compute the gradient of the optimal Sobolev critic on a 2D grid as given in Equation 9 (using numerical evaluation of the CDF and finite difference for the evaluation of the partial derivatives). We plot in Figure 2(b) the density functions of $\mathbb{P}$ and $\mathbb{Q}$ as well as the vector field of the gradient of the optimal Sobolev critic. As discussed in Section A.1, we see that the gradient of the critic (wrt to the input), defines on the support of $\mu = \frac{\mathbb{P}+\mathbb{Q}}{2}$ a transportation plan for moving the distribution mass from $\mathbb{Q}$ to $\mathbb{P}$.

## 5  SOBOLEV GAN

Now we turn to the problem of learning GANs with Sobolev IPM. Given the "real distribution" $\mathbb{P}_r \in \mathscr{P}(\mathcal{X})$, our goal is to learn a generator $g_\theta : \mathcal{Z} \subset \mathbb{R}^{n_z} \to \mathcal{X}$, such that for $z \sim p_z$, the distribution of $g_\theta(z)$ is close to the real data distribution $\mathbb{P}_r$, where $p_z$ is a fixed distribution on $\mathcal{Z}$ (for instance $z \sim \mathscr{N}(0, I_{n_z})$). We note $\mathbb{Q}_\theta$ for the "fake distribution" of $g_\theta(z), z \sim p_z$. Consider $\{x_i, i = 1 \ldots N\} \sim \mathbb{P}_r, \{z_i, i = 1 \ldots N\} \sim \mathscr{N}(0, I_{n_z})$, and $\{\tilde{x}_i, i = 1 \ldots N\} \sim \mu$. We consider these choices for $\mu$:

1. $\mu = \frac{\mathbb{P}_r + \mathbb{Q}_\theta}{2}$ i.e $\tilde{x} \sim \mathbb{P}_r$ or $\tilde{x} = g_\theta(z), z \sim p_z$ with equal probability $\frac{1}{2}$.

2. $\mu_{GP}$ is the implicit distribution defined by the interpolation lines between $\mathbb{P}_r$ and $\mathbb{Q}_\theta$ as in (Gulrajani et al., 2017) i.e : $\tilde{x} = ux + (1-u)y, x \sim \mathbb{P}_r, y = g_\theta(z), z \sim p_z$ and $u \sim \text{Unif}[0, 1]$.

Sobolev GAN can be written as follows:

$$\min_{g_\theta} \sup_{f_p, \frac{1}{N}\sum_{i=1}^{N}\|\nabla_x f_p(\tilde{x}_i)\|^2 = 1} \hat{\mathscr{E}}(f_p, g_\theta) = \frac{1}{N}\sum_{i=1}^{N} f_p(x_i) - \frac{1}{N}\sum_{i=1}^{N} f_p(g_\theta(z_i))$$

For any choice of the parametric function class $\mathscr{H}_p$, note the constraint by $\hat{\Omega}_S(f_p, g_\theta) = \frac{1}{N}\sum_{i=1}^{N}\|\nabla_x f_p(\tilde{x}_i)\|^2$. For example if $\mu = \frac{\mathbb{P}_r+\mathbb{Q}_\theta}{2}$, $\hat{\Omega}_S(f_p, g_\theta) = \frac{1}{2N}\sum_{i=1}^{N}\|\nabla_x f_p(x_i)\|^2 +$

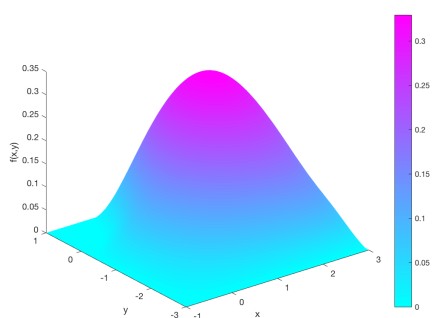
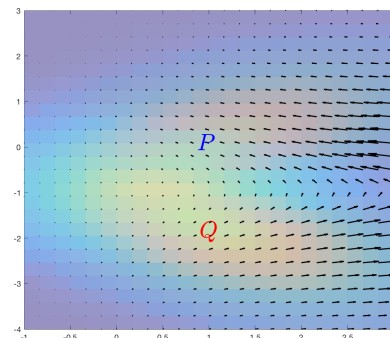

(a) Numerical solution of the PDE satisfied by the optimal Sobolev critic.

(b) Optimal Sobolev Transport Vector Field $\nabla_x f^*(x)$ (arrows are the vector field $\nabla_x f^*(x)$ evaluated on the 2D grid. Magnitude of arrows was rescaled for visualization.)

Figure 2: Numerical solution of the PDE satisfied by the optimal Sobolev critic and the transportation Plan induced by the gradient of Sobolev critic. The *gradient of the critic* (wrt to the input), defines on the support of $\mu = \frac{\mathbb{P}+\mathbb{Q}}{2}$ a *transportation plan for moving the distribution mass from $\mathbb{Q}$ to $\mathbb{P}$*. For a theoretical analysis of this transportation plan and its relation to Fokker-Planck diffusion the reader is invited to check Appendix A.

$\frac{1}{2N}\sum_{i=1}^{N}\|\nabla_x f_p(g_\theta(z_i))\|^2$. Note that, since the optimal theoretical critic is achieved on the sphere, we impose a sphere constraint rather than a ball constraint. Similar to (Mroueh & Sercu, 2017) we define the Augmented Lagrangian corresponding to Sobolev GAN objective and constraint

$$\mathcal{L}_S(p,\theta,\lambda) = \hat{\mathscr{E}}(f_p,g_\theta) + \lambda(1-\hat{\Omega}_S(f_p,g_\theta)) - \frac{\rho}{2}(\hat{\Omega}_S(f_p,g_\theta)-1)^2 \qquad (10)$$

where $\lambda$ is the Lagrange multiplier and $\rho > 0$ is the quadratic penalty weight. We alternate between optimizing the critic and the generator. We impose the constraint when training the critic only. Given $\theta$, we solve $\max_p \min_\lambda \mathcal{L}_S(p,\theta,\lambda)$, for training the critic. Then given the critic parameters $p$ we optimize the generator weights $\theta$ to minimize the objective $\min_\theta \hat{\mathscr{E}}(f_p,g_\theta)$. See Algorithm 1.

---

**Algorithm 1** Sobolev GAN

**Input:** $\rho$ penalty weight, $\eta$ Learning rate, $n_c$ number of iterations for training the critic, N batch size

**Initialize** $p,\theta,\lambda = 0$

**repeat**

    **for** $j = 1$ **to** $n_c$ **do**

        Sample a minibatch $x_i, i = 1\dots N, x_i \sim \mathbb{P}_r$

        Sample a minibatch $z_i, i = 1\dots N, z_i \sim p_z$

        $(g_p, g_\lambda) \leftarrow (\nabla_p \mathcal{L}_S, \nabla_\lambda \mathcal{L}_S)(p,\theta,\lambda)$

        $p \leftarrow p + \eta$ ADAM $(p, g_p)$

        $\lambda \leftarrow \lambda - \rho g_\lambda$ {SGD rule on $\lambda$ with learning rate $\rho$}

    **end for**

    Sample $z_i, i = 1\dots N, z_i \sim p_z$

    $d_\theta \leftarrow \nabla_\theta \hat{\mathscr{E}}(f_p,g_\theta) = -\nabla_\theta \frac{1}{N}\sum_{i=1}^{N} f_p(g_\theta(z_i))$

    $\theta \leftarrow \theta - \eta$ ADAM $(\theta, d_\theta)$

**until** $\theta$ converges

---

**Remark 1.** *Note that in Algorithm 1, we obtain a biased estimate since we are using same samples for the cost function and the constraint, but the incurred bias can be shown to be small and vanishing as the number of samples increases as shown and justified in (Shivaswamy & Jebara, 2010).*

**Relation to WGAN-GP.** WGAN-GP can be written as follows:

$$\min_{g_\theta} \sup_{f, \|\nabla_x f_p(\tilde{x}_i)\|=1, \tilde{x}_i \sim \mu_{GP}} \hat{\mathscr{E}}(f_p, g_\theta) = \frac{1}{N} \sum_{i=1}^{N} f_p(x_i) - \frac{1}{N} \sum_{i=1}^{N} f_p(g_\theta(z_i))$$

The main difference between WGAN-GP and our setting, is that WGAN-GP enforces *pointwise constraints* on points drawn from $\mu = \mu_{GP}$ via a point-wise quadratic penalty ($\hat{\mathscr{E}}(f_p, g_\theta) - \lambda \sum_{i=1}^{N}(1 - \|\nabla_x f(\tilde{x}_i)\|)^2$) while we enforce that constraint on average as a Sobolev norm, allowing us the coordinate weighted conditional CDF interpretation of the IPM.

## 6 APPLICATIONS OF SOBOLEV GAN

Sobolev IPM has two important properties; The first stems from the *conditioning* built in to the metric through the weighted conditional CDF interpretation. The second stems from the *diffusion* properties that the critic of Sobolev IPM satisfies (Appendix A) that has theoretical and practical ties to the Laplacian regularizer and diffusion on manifolds used in semi-supervised learning (Belkin et al., 2006).

In this Section, we exploit those two important properties in two applications of Sobolev GAN: Text generation and semi-supervised learning. First in *text generation*, which can be seen as a discrete sequence generation, Sobolev GAN (and WGAN-GP) enable training GANs without need to do explicit brute-force conditioning. We attribute this to the built-in conditioning in Sobolev IPM (for the sequence aspect) and to the CDF matching (for the discrete aspect). Secondly using GANs in semi-supervised learning is a promising avenue for learning using unlabeled data. We show that a variant of Sobolev GAN can achieve strong SSL results on the CIFAR-10 dataset, without the need of any form of activation normalization in the networks or any extra ad hoc tricks.

### 6.1 TEXT GENERATION WITH SOBOLEV GAN

In this Section, we present an empirical study of Sobolev GAN in character level text generation. Our empirical study on end to end training of character-level GAN for text generation is articulated on four dimensions (**loss, critic, generator, $\mu$**). (1) the loss used (**GP**: WGAN-GP (Gulrajani et al., 2017), **S**: Sobolev or **F**: Fisher) (2) the architecture of the critic (Resnets or RNN) (3) the architecture of the generator (Resnets or RNN or RNN with curriculum learning) (4) the sampling distribution $\mu$ in the constraint.

**Text Generation Experiments.** We train a character-level GAN on Google Billion Word dataset and follow the same experimental setup used in (Gulrajani et al., 2017). The generated sequence length is 32 and the evaluation is based on Jensen-Shannon divergence on empirical 4-gram probabilities (JS-4) of validation data and generated data. JS-4 may not be an ideal evaluation criteria, but it is a reasonable metric for current character-level GAN results, which is still far from generating meaningful sentences.

**Annealed Smoothing of discrete $\mathbb{P}_r$ in the constraint $\mu$.** Since the generator distribution will always be defined on a continuous space, we can replace the discrete "real" distribution $\mathbb{P}_r$ with a smoothed version (Gaussian kernel smoothing) $\mathbb{P}_r \star \mathscr{N}(0, \sigma^2 I_d)$. This corresponds to doing the following sampling for $\mathbb{P}_r : x + \xi, x \sim \mathbb{P}_r$, and $\xi \sim \mathscr{N}(0, \sigma^2 I_d)$. Note that we only inject noise to the "real" distribution with the goal of smoothing the support of the discrete distribution, as opposed to instance noise on both "real" and "fake" to stabilize the training, as introduced in (Kaae Sønderby et al., 2017; Arjovsky & Bottou, 2017). As it is common in optimization by continuation (Mobahi & III, 2015), we also anneal the noise level $\sigma$ as the training progresses on a linear schedule.

**Sobolev GAN versus WGAN-GP with Resnets.** In this setting, we compare (WGAN-GP,G=Resnet,D=Resnet,$\mu = \mu_{GP}$) to (Sobolev,G=Resnet,D=Resnet,$\mu$) where $\mu$ is one of: (1) $\mu_{GP}$, (2) the noise smoothed $\mu_s(\sigma) = \frac{\mathbb{P}_r \star \mathscr{N}(0, \sigma^2 I_d) + \mathbb{Q}_\theta}{2}$ or (3) noise smoothed with annealing $\mu_s^a(\sigma_0)$ with $\sigma_0$ the initial noise level. We use the same architectures of Resnet with 1D convolution for the critic and the generator as in (Gulrajani et al., 2017) (4 resnet blocks with hidden layer size of 512). In order to implement the noise smoothing we transform the data into one-hot vectors. Each one hot vector $x$ is transformed to a probability vector $p$ with 0.9 replacing the one and $0.1/(dict_{size} - 1)$ replacing the zero. We then sample $\epsilon$ from a Gaussian distribution $\mathscr{N}(0, \sigma^2)$, and

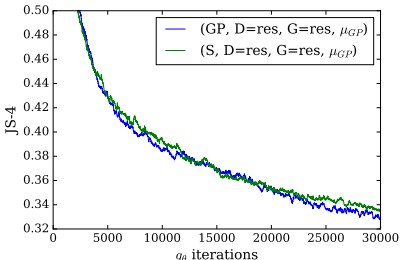
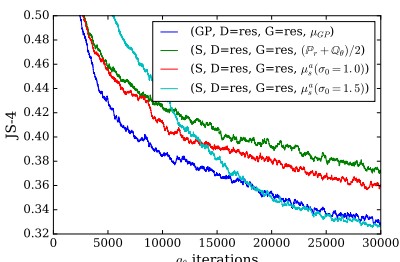
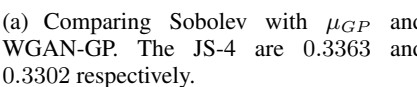

(a) Comparing Sobolev with $\mu_{GP}$ and WGAN-GP. The JS-4 are 0.3363 and 0.3302 respectively.

(b) Comparing Sobolev with different $\mu$ dominant measures and WGAN-GP. The JS-4 of $\mu_s^a(\sigma_0 = 1.5)$ is 0.3268.

Figure 3: Result of Sobolev GAN for various dominating measure $\mu$, for resnets as architectures of the critic and the generator.

use softmax to normalize $\log p + \epsilon$. We use algorithm 1 for Sobolev GAN and fix the learning rate to $10^{-4}$ and $\rho$ to $10^{-5}$. The noise level $\sigma$ was annealed following a linear schedule starting from an initial noise level $\sigma_0$ (at iteration $i$, $\sigma_i = \sigma_0(1 - \frac{i}{Maxiter})$, Maxiter=30K). For WGAN-GP we used the open source implementation with the penalty $\lambda = 10$ as in (Gulrajani et al., 2017). Results are given in Figure 3(a) for the JS-4 evaluation of both WGAN-GP and Sobolev GAN for $\mu = \mu_{GP}$. In Figure 3(b) we show the JS-4 evaluation of Sobolev GAN with the annealed noise smoothing $\mu_s^a(\sigma_0)$, for various values of the initial noise level $\sigma_0$. We see that the training succeeds in both cases. Sobolev GAN achieves slightly better results than WGAN-GP for the annealing that starts with high noise level $\sigma_0 = 1.5$. We note that without smoothing and annealing i.e using $\mu = \frac{\mathbb{P}_r + \mathbb{Q}_\theta}{2}$, Sobolev GAN is behind. Annealed smoothing of $\mathbb{P}_r$, helps the training as the real distribution is slowly going from a continuous distribution to a discrete distribution. See Appendix C (Figure 6) for a comparison between annealed and non annealed smoothing.

We give in Appendix C a comparison of WGAN-GP and Sobolev GAN for a Resnet generator architecture and an RNN critic. The RNN has degraded performance due to optimization difficulties.

**Fisher GAN Curriculum Conditioning versus Sobolev GAN: Explicit versus Implicit conditioning.** We analyze how Fisher GAN behaves under different architectures of generators and critics. We first fix the generator to be ResNet. We study 3 different architectures of critics: ResNet, GRU (we follow the experimental setup from (Press et al., 2017)), and hybrid ResNet+GRU (Reed et al., 2016). We notice that RNN is unstable, we need to clip the gradient values of critics in $[-0.5, 0.5]$, and the gradient of the Lagrange multiplier $\lambda_F$ to $[-10^4, 10^4]$. We fix $\rho_F = 10^{-7}$ and we use $\mu = \mu_{GP}$. We search the value for the learning rate in $[10^{-5}, 10^{-4}]$. We see that for $\mu = \mu_{GP}$ and $G$ = Resnet for various critic architectures, Fisher GAN fails at the task of text generation (Figure 4 a-c). Nevertheless, when using RNN critics (Fig 4 b, c) a marginal improvement happens over the fully collapsed state when using a resnet critic (Fig 4 a). We hypothesize that RNN critics enable some conditioning and factoring of the distribution, which is lacking in Fisher IPM.

Finally Figure 4 (d) shows the result of training with recurrent generator and critic. We follow (Press et al., 2017) in terms of GRU architecture, but differ by using Fisher GAN rather than WGAN-GP. We use $\mu = \frac{\mathbb{P}_r + \mathbb{Q}_\theta}{2}$ i.e. without annealed noise smoothing. We train (F, D=RNN,G=RNN,$\frac{\mathbb{P}_r + \mathbb{Q}_\theta}{2}$) using curriculum conditioning of the generator for all lengths $\ell$ as done in (Press et al., 2017): the generator is conditioned on $32 - \ell$ characters and predicts the $\ell$ remaining characters. We increment $\ell = 1$ to 32 on a regular schedule (every 15k updates). JS-4 is only computed when $\ell > 4$. We see in Figure 4 that under curriculum conditioning with recurrent critics and generators, the training of Fisher GAN succeeds and reaches similar levels of Sobolev GAN (and WGAN-GP). Note that the need of this *explicit brute force conditioning* for Fisher GAN, highlights the *implicit conditioning* induced by Sobolev GAN via the gradient regularizer, without the need for curriculum conditioning.

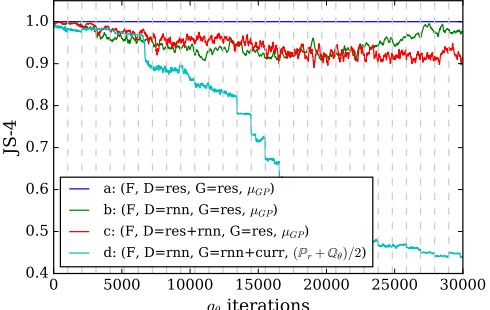

Figure 4: Fisher GAN with different architectures for critics: (a-c) We see that for $\mu = \mu_{GP}$ and $G = $ Resnet for various critic architectures, Fisher GAN fails at the task of text generation. We notice small improvements for RNN critics (b-c) due to the conditioning and factoring of the distribution. (d) Fisher GAN with recurrent generator and critic, trained on a curriculum conditioning for increasing lengths $\ell$, increments indicated by gridlines. In this curriculum conditioning setup, with recurrent critics and generators, the training of Fisher GAN succeeds and reaches similar levels of Sobolev GAN (and WGAN-GP). It is important to note that by doing this *explicit curriculum conditioning* for Fisher GAN, we highlight the *implicit conditioning* induced by Sobolev GAN, via the gradient regularizer.

## 6.2 SEMI-SUPERVISED LEARNING WITH SOBOLEV GAN

A proper and promising framework for evaluating GANs consists in using it as a regularizer in the semi-supervised learning setting (Salimans et al., 2016; Dumoulin et al., 2017; Kumar et al., 2017). As mentioned before, the Sobolev norm as a regularizer for the Sobolev IPM draws connections with the Laplacian regularization in manifold learning (Belkin et al., 2006). In the Laplacian framework of semi-supervised learning, the classifier satisfies a smoothness constraint imposed by controlling its Sobolev norm: $\int_{\mathcal{X}} \|\nabla_x f(x)\|^2 \mu^2(x) dx$ (Alaoui et al., 2016). In this Section, we present a variant of Sobolev GAN that achieves competitive performance in semi-supervised learning on the CIFAR-10 dataset Krizhevsky & Hinton (2009) without using any internal activation normalization in the critic, such as batch normalization (BN) (Ioffe & Szegedy, 2015), layer normalization (LN) (Ba et al., 2016), or weight normalization (Salimans & Kingma, 2016).

In this setting, a convolutional neural network $\Phi_\omega : \mathcal{X} \to \mathbb{R}^m$ is shared between the cross entropy (CE) training of a $K$-class classifier ($S \in \mathbb{R}^{K \times m}$) and the critic of GAN (See Figure 5). We have the following training equations for the (critic + classifer) and the generator:

$$\text{Critic + Classifier:} \quad \max_{S, \Phi_\omega, f} \mathcal{L}_D = \mathcal{L}_{\text{alm}}^{\text{GAN}}(f, g_\theta) - \lambda_{CE} \sum_{(x,y) \in \text{lab}} CE(p(y|x), y) \tag{11}$$

$$\text{Generator:} \quad \max_\theta \mathcal{L}_G = \hat{\mathscr{E}}(f, g_\theta) \tag{12}$$

where the main IPM objective with $N$ samples: $\hat{\mathscr{E}}(f, g_\theta) = \frac{1}{N} \left( \sum_{x \in \text{unl}} f(x) - \sum_{z \sim p_z} f(g_\theta(z)) \right)$.

Following (Mroueh & Sercu, 2017) we use the following "$K + 1$ parametrization" for the critic (See Figure 5) :

$$f(x) = \underbrace{\sum_{y=1}^{K} p(y|x) \langle S_y, \Phi_\omega(x) \rangle}_{\boldsymbol{f_+}: \text{"real" critic}} - \underbrace{\langle v, \Phi_\omega(x) \rangle}_{\boldsymbol{f_-}: \text{"fake" critic}}$$

Note that $p(y|x) = \text{Softmax}(\langle S, \Phi_\omega(x) \rangle)_y$ appears both in the critic formulation and in the Cross-Entropy term in Equation (11). Intuitively this critic uses the $K$ class directions of the classifier $S_y$ to define the "real" direction, which competes with another $K+1^{\text{th}}$ direction $v$ that indicates fake samples. This parametrization adapts the idea of (Salimans et al., 2016), which was formulated specifically for the classic KL / JSD based GANs, to IPM-based GANs. We saw consistently better results with the $K + 1$ formulation over the regular formulation where the classification layer $S$

doesn't interact with the critic direction $v$. We also note that when applying a gradient penalty based constraint (either WGAN-GP or Sobolev) on the full critic $f = f_+ - f_-$, it is impossible for the network to fit even the small labeled training set (underfitting), causing bad SSL performance. This leads us to the formulation below, where we apply the Sobolev constraint only on $\boldsymbol{f}_-$. Throughout this Section we fix $\mu = \frac{\mathbb{P}_r + \mathbb{Q}_\theta}{2}$.

We propose the following two schemes for constraining the K+1 critic $f(x) = f_+(x) - f_-(x)$:

1) **Fisher constraint on the critic**: We restrict the critic to the following set:

$$f \in \left\{ f = f_+ - f_-, \ \hat{\Omega}_F(f, g_\theta) = \frac{1}{2N} \left( \sum_{x \in \text{unl}} f^2(x) + \sum_{z \sim p_z} f^2(g_\theta(z)) \right) = 1 \right\}.$$

This constraint translates to the following ALM objective in Equation (11):

$$\mathcal{L}_{\text{alm}}^{\text{GAN}}(f, g_\theta) = \hat{\mathscr{E}}(f, g_\theta) + \lambda_F(1 - \hat{\Omega}_F(f, g_\theta)) - \frac{\rho_F}{2}(\hat{\Omega}_F(f, g_\theta) - 1)^2,$$

where the Fisher constraint ensures the stability of the training through an implicit whitened mean matching (Mroueh & Sercu, 2017).

2) **Fisher+Sobolev constraint:** We impose 2 constraints on the critic: Fisher on $\boldsymbol{f}$ & Sobolev on $\boldsymbol{f}_-$

$$f \in \left\{ f = f_+ - f_-, \ \hat{\Omega}_F(\boldsymbol{f}, g_\theta) = 1 \text{ and } \hat{\Omega}_S(\boldsymbol{f}_-, g_\theta) = 1 \right\},$$

where $\hat{\Omega}_S(\boldsymbol{f}_-, g_\theta) = \frac{1}{2N} \left( \sum_{x \in \text{unl}} \|\nabla_x \boldsymbol{f}_-(x)\|^2 + \sum_{z \sim p_z} \|\nabla_x \boldsymbol{f}_-(g_\theta(z))\|^2 \right)$.

This constraint translates to the following ALM in Equation (11):

$$\mathcal{L}_{\text{alm}}^{\text{GAN}}(f, g_\theta) = \hat{\mathscr{E}}(f, g_\theta) + \lambda_F(1 - \hat{\Omega}_F(\boldsymbol{f}, g_\theta)) + \lambda_S(1 - \hat{\Omega}_S(\boldsymbol{f}_-, g_\theta))$$
$$- \frac{\rho_F}{2}(\hat{\Omega}_F(\boldsymbol{f}, g_\theta) - 1)^2 - \frac{\rho_S}{2}(\hat{\Omega}_S(\boldsymbol{f}_-, g_\theta) - 1)^2.$$

Note that the fisher constraint on $\boldsymbol{f}$ ensures the stability of the training, and the Sobolev constraints on the "fake" critic $\boldsymbol{f}_-$ enforces smoothness of the "fake" critic and thus the shared CNN $\Phi_\omega(x)$. This is related to the classic Laplacian regularization in semi-supervised learning (Belkin et al., 2006).

Table 2 shows results of SSL on CIFAR-10 comparing the two proposed formulations. Similar to the standard procedure in other GAN papers, we do hyperparameter and model selection on the validation set. We present baselines with a similar model architecture and leave out results with significantly larger convnets. G and D architectures and hyperparameters are in Appendix D. $\Phi_\omega$ is similar to (Salimans et al., 2016; Dumoulin et al., 2017; Mroueh & Sercu, 2017) in architecture, but note that we do not use any batch, layer, or weight normalization yet obtain strong competitive accuracies. We hypothesize that we don't need any normalization in the critic, because of the implicit whitening of the feature maps introduced by the Fisher and Sobolev constraints as explained in (Mroueh & Sercu, 2017).

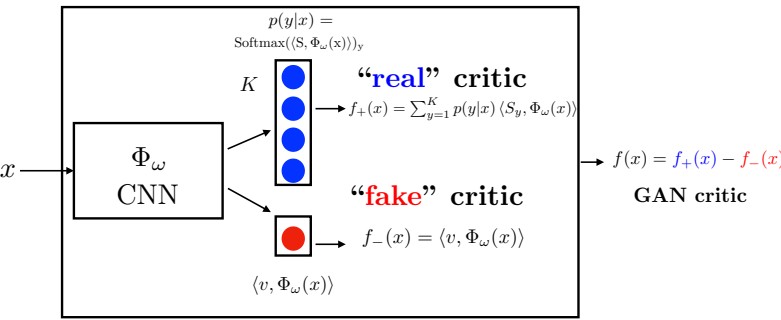

Figure 5: "K+1" parametrization of the critic for semi-supervised learning.

Table 2: CIFAR-10 error rates for varying number of labeled samples in the training set. Mean and standard deviation computed over 5 runs. We only use the $K + 1$ formulation of the critic. Note that we achieve strong SSL performance without any additional tricks, and even though the critic does not have any batch, layer or weight normalization. Baselines with * use either additional models like PixelCNN, or do data augmentation (translations and flips), or use a much larger model, either of which gives an advantage over our plain simple training method. † is the result we achieved in our experimental setup under the same conditions but without "K+1" critic (see Appendix D), since (Gulrajani et al., 2017) does not have SSL results.

| Number of labeled examples | 1000 | 2000 | 4000 | 8000 |
|---|---|---|---|---|
| Model | | Misclassification rate | | |
| CatGAN (Springenberg, 2015) | | | 19.58 | |
| FM (Salimans et al., 2016) | $21.83 \pm 2.01$ | $19.61 \pm 2.09$ | $18.63 \pm 2.32$ | $17.72 \pm 1.82$ |
| ALI (Dumoulin et al., 2017) | $19.98 \pm 0.3$ | $19.09 \pm 0.15$ | $17.99 \pm 0.54$ | $17.05 \pm 0.50$ |
| Tangents Reg (Kumar et al., 2017) | $20.06 \pm 0.5$ | | $16.78 \pm 0.6$ | |
| Π-model (Laine & Aila, 2016) * | | | $16.55 \pm 0.29$ | |
| VAT (Miyato et al., 2017) | | | 14.87 | |
| Bad Gan (Dai et al., 2017) * | | | $14.41 \pm 0.30$ | |
| VAT+EntMin+Large (Miyato et al., 2017) * | | | 13.15 | |
| Sajjadi (Sajjadi et al., 2016) * | | | 11.29 | |
| WGAN-GP (Gulrajani et al., 2017) † | $44.85 \pm 0.28$ | $37.62 \pm 0.56$ | $32.66 \pm 0.48$ | $30.38 \pm 0.22$ |
| Fisher, layer norm (Mroueh & Sercu, 2017) | $19.74 \pm 0.21$ | $17.87 \pm 0.38$ | $16.13 \pm 0.53$ | $14.81 \pm 0.16$ |
| Fisher, no norm (Mroueh & Sercu, 2017) | $21.49 \pm 0.18$ | $19.20 \pm 0.46$ | $17.30 \pm 0.30$ | $15.57 \pm 0.33$ |
| Sobolev + Fisher, no norm (This Work) | $20.14 \pm 0.21$ | $17.38 \pm 0.10$ | $15.77 \pm 0.19$ | $14.20 \pm 0.08$ |

## 7 CONCLUSION

We introduced the Sobolev IPM and showed that it amounts to a comparison between weighted (coordinate-wise) CDFs. We presented an ALM algorithm for training Sobolev GAN. The intrinsic conditioning implied by the Sobolev IPM explains the success of gradient regularization in Sobolev GAN and WGAN-GP on discrete sequence data, and particularly in text generation. We highlighted the important tradeoffs between the implicit conditioning introduced by the gradient regularizer in Sobolev IPM, and the explicit conditioning of Fisher IPM via recurrent critics and generators in conjunction with the curriculum conditioning. Both approaches succeed in text generation. We showed that Sobolev GAN achieves competitive semi-supervised learning results without the need of any normalization, thanks to the smoothness induced by the gradient regularizer. We think the Sobolev IPM point of view will open the door for designing new regularizers that induce different types of conditioning for general structured/discrete/graph data beyond sequences.

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

# A  THEORY: APPROXIMATION AND TRANSPORT INTERPRETATION

In this Section we present the theoretical properties of Sobolev IPM and how it relates to distributions transport theory and other known metrics between distributions, notably the Stein distance.

## A.1  DISTRIBUTION TRANSPORT PERSPECTIVE ON SOBOLEV IPM

In this Section, we characterize the optimal critic of the Sobolev IPM as a solution of a non linear PDE. The solution of the variational problem of the Sobolev IPM satisfies a non linear PDE that can be derived using standard tools from calculus of variations (Ekeland & Turnbull, 1983; Alaoui et al., 2016).

**Theorem 3** (PDE satisfied by the Sobolev Critic). *The optimal critic of Sobolev IPM $f^*$ satisfies the following PDE:*

$$\Delta f^*(x) + \langle \nabla_x \log \mu(x), \nabla_x f^*(x) \rangle + \frac{\mathbb{P}(x) - \mathbb{Q}(x)}{\mathcal{S}_\mu(\mathbb{P}, \mathbb{Q})\mu(x)} = 0. \tag{13}$$

Define the Stein Operator: $T(\mu)\vec{g}(x) = \frac{1}{2}\Big( \langle \nabla_x \log(\mu(x)), \vec{g}(x) \rangle + div(\vec{g}(x)) \Big)$. Hence we have the following Transport Equation of $\mathbb{P}$ to $\mathbb{Q}$:

$$\mathbb{Q}(x) = \mathbb{P}(x) + 2\mathcal{S}_\mu(\mathbb{P}, \mathbb{Q})\mu(x)T(\mu)\nabla_x f^*(x).$$

Recall the definition of Stein Discrepancy :

$$\mathbb{S}(\mathbb{Q}, \mu) = \sup_{\vec{g}} |\mathbb{E}_{x \sim \mathbb{Q}}[T(\mu)\vec{g}(x)]|, \vec{g} : \mathcal{X} \to \mathbb{R}^d.$$

**Theorem 4** (Sobolev and Stein Discrepanices). *The following inequality holds true:*

$$\left| \mathbb{E}_{x \sim \mathbb{Q}}\left[ \frac{\mathbb{Q}(x) - \mathbb{P}(x)}{\mu(x)} \right] \right| \leq 2 \underbrace{\mathbb{S}(\mathbb{Q}, \mu)}_{\textit{Stein Good fitness of the model } \mathbb{Q} \textit{ w.r.t to } \mu} \underbrace{\mathcal{S}_\mu(\mathbb{P}, \mathbb{Q})}_{\textit{Sobolev Distance}} \tag{14}$$

Consider for example $\mu = \mathbb{P}$, and sequence $\mathbb{Q}_n$. If the Sobolev distance goes $\mathcal{S}_\mathbb{P}(\mathbb{P}, \mathbb{Q}_n) \to 0$, the ratio $r_n(x) = \frac{\mathbb{Q}_n(x)}{\mathbb{P}(x)}$ converges in expectation (w.r.t to $\mathbb{Q}$) to 1. The speed of the convergence is given by the Stein Discrepancy $\mathbb{S}(\mathbb{Q}_n, \mathbb{P})$.

**Relation to Fokker-Planck Diffusion Equation and Particles dynamics.** Note that PDE satisifed by the Sobolev critic given in Equation (13) can be equivalently written:

$$\frac{\mathbb{P} - \mathbb{Q}}{\mathcal{S}_\mu(\mathbb{P}, \mathbb{Q})} = -\text{div}(\mu(x)\nabla_x f^*(x)), \tag{15}$$

written in this form, we draw a connection with the Fokker-Planck Equation for the evolution of a density function $q_t$ that is the density of particles $X_t \in \mathbb{R}^d$ evolving with a drift (a velocity field) $V(x, t) : \mathcal{X} \times [0, \infty[ \to \mathbb{R}^d$:

$$dX_t = V(X_t, t)dt, \text{ where the density of } X_0 \text{ is given by } q_0(x) = \mathbb{Q}(x),$$

The Fokker-Planck Equation states that the evolution of the particles density $q_t$ satisfies:

$$\frac{dq_t}{dt}(x) = -\text{div}(q_t(x)V(x, t)) \tag{16}$$

Comparing Equation (15) and Equation (16), we identify then the gradient of Sobolev critic as a drift. This suggests that one can define "Sobolev descent" as the evolution of particles along the gradient flow:

$$dX_t = \nabla_x f_t^*(X_t)dt, \text{ where the density of } X_0 \text{ is given by } q_0(x) = \mathbb{Q}(x),$$

where $f_t^*$ is the Sobolev critic between $q_t$ and $\mathbb{P}$. One can show that the limit distribution of the particles is $\mathbb{P}$. The analysis of "Sobolev descent" and its relation to Stein Descent (Liu & Wang, 2016; Liu, 2017) is beyond the scope of this paper and will be studied in a separate work. Hence we see that the gradient of the Sobolev critic defines a transportation plan to move particles whose distribution is $\mathbb{Q}$ to particles whose distribution is $\mathbb{P}$ (See Figure 2). This highlights the role of the gradient of the critic in the context of GAN training in term of transporting the distribution of the generator to the real distribution.

## B  PROOFS

*Proof of Theorem 2.* Let $F_{\mathbb{P}}$ and $F_{\mathbb{Q}}$, be *the cumulative distribution functions* of $\mathbb{P}$ and $\mathbb{Q}$ respectively. We have:

$$\mathbb{P}(x) = \frac{\partial^d}{\partial x_1 \dots \partial x_d} F_{\mathbb{P}}(x), \tag{17}$$

We note $D = \frac{\partial^d}{\partial x_1 \dots \partial x_d}$, and $D^{-i} = \frac{\partial^{d-1}}{\partial x_1 \dots \partial x_{i-1} \partial x_{i+1} \dots \partial x_d}$, for $i = 1 \dots d$.

$D^{-i}$ computes the $d-1$ partial derivative excluding the variable $i$.
In the following we assume that $F_{\mathbb{P}}$, and $F_{\mathbb{Q}}$ and its $d$ derivatives exist and are continuous meaning that $F_{\mathbb{P}}$ and $F_{\mathbb{Q}} \in C^d(\mathcal{X})$. The objective function in Equation (3) can be written as follows:

$$
\begin{aligned}
\mathbb{E}_{x \sim \mathbb{P}} f(x) - \mathbb{E}_{x \sim \mathbb{Q}} f(x) &= \int_{\mathcal{X}} f(x) D\Big(F_{\mathbb{P}}(x) - F_{\mathbb{Q}}(x)\Big) dx \\
&= \int_{\mathcal{X}} f(x) \frac{\partial}{\partial x_i} D^{-i}(F_{\mathbb{P}}(x) - F_{\mathbb{Q}}(x)) dx \\
&\quad \text{(for any } i, \text{ since } F_{\mathbb{P}} \text{ and } F_{\mathbb{Q}} \in C^d(\mathcal{X})) \\
&= -\int_{\mathcal{X}} \frac{\partial f}{\partial x_i} D^{-i}(F_{\mathbb{P}}(x) - F_{\mathbb{Q}}(x)) dx \\
&\quad (f \text{ vanishes at the boundary in } W_0^{1,2}(\mathcal{X}, \mu) )
\end{aligned}
$$

Let $D^- = (D^{-1}, \dots, D^{-d})$ it follows that:

$$
\begin{aligned}
\mathbb{E}_{x \sim \mathbb{P}} f(x) - \mathbb{E}_{x \sim \mathbb{Q}} f(x) &= \frac{1}{d} \sum_{i=1}^{d} \int_{\mathcal{X}} \frac{\partial f}{\partial x_i} D^{-i}(F_{\mathbb{Q}}(x) - F_{\mathbb{P}}(x)) dx \\
&= \frac{1}{d} \int_{\mathcal{X}} \left\langle \nabla_x f(x), D^-(F_{\mathbb{Q}}(x) - F_{\mathbb{P}}(x)) \right\rangle_{\mathbb{R}^d} dx
\end{aligned} \tag{18}
$$

Let us define $\mathscr{L}_2(\mathcal{X}, \mu)^{\otimes d}$ the space of measurable functions from $\mathcal{X} \to \mathbb{R}^d$. For $g, h \in \mathscr{L}_2(\mathcal{X}, \mu)^{\otimes d}$ the dot product is defined as follows:

$$\langle g, h \rangle_{\mathscr{L}_2(\mathcal{X}, \mu)^{\otimes d}} = \int_{\mathcal{X}} \langle g(x), h(x) \rangle_{\mathbb{R}^d} \, \mu(x) dx$$

and the norm is given :

$$\|g\|_{\mathscr{L}_2(\mathcal{X}, \mu)^{\otimes d}} = \int_X \|g\|_{\mathbb{R}^d}^2 \, \mu(x) dx.$$

We can write the objective in Equation (18) in term of the dot product in $\mathscr{L}_2(\mathcal{X}, \mu)^{\otimes d}$ :

$$\mathbb{E}_{x \sim \mathbb{P}} f(x) - \mathbb{E}_{x \sim \mathbb{Q}} f(x) = \frac{1}{d} \left\langle \nabla_x f, \frac{D^-(F_{\mathbb{Q}} - F_{\mathbb{P}})}{\mu} \right\rangle_{\mathscr{L}_2(\mathcal{X}, \mu)^{\otimes d}}. \tag{19}$$

On the other hand the constraint in Equation (3) can be written in terms of the norm in $\mathscr{L}_2(\mathcal{X}, \mu)^{\otimes d}$:

$$\|f\|_{W_0^{1,2}(\mathcal{X}, \mu)} = \|\nabla_x f\|_{\mathscr{L}_2(\mathcal{X}, \mu)^{\otimes d}} \tag{20}$$

Replacing the objective and constraint given in Equations (19) and (20) in Equation (3), we obtain:

$$
\begin{aligned}
\mathcal{S}(\mathbb{P}, \mathbb{Q}) &= \frac{1}{d} \sup_{f, \|\nabla_x f\|_{\mathscr{L}_2(\mathcal{X},\mu)^{\otimes d}} \leq 1} \left\langle \nabla_x f, \frac{D^-(F_{\mathbb{Q}} - F_{\mathbb{P}})}{\mu} \right\rangle_{\mathscr{L}_2(\mathcal{X},\mu)^{\otimes d}} \\
&= \frac{1}{d} \sup_{g \in \mathscr{L}_2(\mathcal{X},\mu)^{\otimes d}, \|g\|_{\mathscr{L}_2(\mathcal{X},\mu)^{\otimes d}} \leq 1} \left\langle g, \frac{D^-(F_{\mathbb{Q}} - F_{\mathbb{P}})}{\mu} \right\rangle_{\mathscr{L}_2(\mathcal{X},\mu)^{\otimes d}} \\
&= \frac{1}{d} \left\| \frac{D^-(F_{\mathbb{Q}} - F_{\mathbb{P}})}{\mu} \right\|_{\mathscr{L}_2(\mathcal{X},\mu)^{\otimes d}}
\end{aligned}
$$

$$
\left( \text{By definition of } \|.\|_{\mathscr{L}_2(\mathcal{X},\mu)^{\otimes d}}, g^* = \frac{D^- F_{\mathbb{Q}}(x) - D^- F_{\mathbb{P}}(x)}{\mu(x)} \frac{1}{\left\| \frac{D^-(F_{\mathbb{Q}} - F_{\mathbb{P}})}{\mu} \right\|_{\mathscr{L}_2(\mathcal{X},\mu)^{\otimes d}}} \right)
$$

$$
= \frac{1}{d} \sqrt{\int_{\mathcal{X}} \frac{\left\| D^- F_{\mathbb{Q}}(x) - D^- F_{\mathbb{P}}(x) \right\|^2}{\mu(x)} dx}.
$$

Hence we find also that the optimal critic $f^*$ satisfies:

$$
\nabla_x f^*(x) = \frac{D^- F_{\mathbb{Q}}(x) - D^- F_{\mathbb{P}}(x)}{\mu(x)} \frac{1}{\left\| \frac{D^-(F_{\mathbb{Q}} - F_{\mathbb{P}})}{\mu} \right\|_{\mathscr{L}_2(\mathcal{X},\mu)^{\otimes d}}}.
$$

$\square$

*Proof of Lemma 1.*

$$
\begin{aligned}
\mathbb{E}_{x \sim \mathbb{P}} f(x) - \mathbb{E}_{x \sim \mathbb{Q}} f(x) &= \frac{1}{d} \int_{\mathcal{X}} \left\langle \nabla_x f(x), D^-(F_{\mathbb{Q}}(x) - F_{\mathbb{P}}(x)) \right\rangle_{\mathbb{R}^d} dx \\
&= \mathcal{S}_\mu(\mathbb{P}, \mathbb{Q}) \int_{\mathcal{X}} \left\langle \nabla_x f(x), \frac{D^-(F_{\mathbb{Q}}(x) - F_{\mathbb{P}}(x))}{\mu(x) d \mathcal{S}_\mu(\mathbb{P}, \mathbb{Q})} \right\rangle_{\mathbb{R}^d} \mu(x) dx \\
&= \mathcal{S}_\mu(\mathbb{P}, \mathbb{Q}) \int_{\mathcal{X}} \left\langle \nabla_x f(x), \nabla_x f^*(x) \right\rangle \mu(x) dx \\
&= \mathcal{S}_\mu(\mathbb{P}, \mathbb{Q}) \left\langle f, f^* \right\rangle_{W_0^{1,2}}
\end{aligned}
$$

Hence we have:

$$
\sup_{f \in \mathscr{H}, \|f\|_{W_0^{1,2}} \leq 1} \mathbb{E}_{x \sim \mathbb{P}} f(x) - \mathbb{E}_{x \sim \mathbb{Q}} f(x) = \mathcal{S}_\mu(\mathbb{P}, \mathbb{Q}) \sup_{f \in \mathscr{H}, \|f\|_{W_0^{1,2}} \leq 1} \left\langle f, f^* \right\rangle_{W_0^{1,2}},
$$

It follows therefore that:

$$
\mathcal{S}_{\mathscr{H}}(\mathbb{P}, \mathbb{Q}) = \mathcal{S}_\mu(\mathbb{P}, \mathbb{Q}) \sup_{f \in \mathscr{H}, \|f\|_{W_0^{1,2}} \leq 1} \left\langle f, f^* \right\rangle_{W_0^{1,2}}
$$

$\square$

We conclude that the Sobolev IPM can be approximated in arbitrary space as long as it has enough capacity to approximate the optimal critic. Interestingly the approximation error is measured now with the Sobolev semi-norm, while in Fisher it was measured with the Lebesgue norm. Approximations with Sobolev Semi-norms are stronger then Lebesgue norms as given by the Poincare inequality ($\|f\|_{\mathscr{L}_2} \leq C \|f\|_{W_0^{1,2}}$), meaning if the error goes to zero in Sobolev sense it also goes to zero in the Lebesgue sense, but the converse is not true.

*Proof of Theorem 3.* The proof follows similar arguments in the proofs of the analysis of Laplacian regularization in semi-supervised learning studied by (Alaoui et al., 2016).

$$
\begin{aligned}
\mathcal{S}_\mu(\mathbb{P}, \mathbb{Q}) &= \sup_{f \in W_0^{1,2}} \left\{ \mathbb{E}_{x \sim \mathbb{P}} [f(x)] - \mathbb{E}_{x \sim \mathbb{Q}} [f(x)] \right\} \\
\text{s.t.} &\qquad \mathbb{E}_{x \sim \mu} \|\nabla f(x)\|_2^2 \leq 1,
\end{aligned} \tag{21}
$$

Note that this problem is convex in $f$ (Ekeland & Turnbull, 1983). Writing the lagrangian for equation (21) we get :

$$L(f, \lambda) = \mathbb{E}_{x \sim \mathbb{P}}[f(x)] - \mathbb{E}_{x \sim \mathbb{Q}}[f(x)] + \frac{\lambda}{2}\Big(1 - \mathbb{E}_{x \sim \mu}\|\nabla_x f(x)\|_2^2\Big)$$

$$= \int_{\mathcal{X}} f(x)\left(\mathbb{P}(x) - \mathbb{Q}(x)\right) dx + \frac{\lambda}{2}\Big(1 - \int_{\mathcal{X}} \|\nabla_x f(x)\|_2^2 \mu(x) dx\Big)$$

$$= \int_{\mathcal{X}} f(x)\,\mu_1(x)\,dx + \frac{\lambda}{2}\Big(1 - \int_{\mathcal{X}} \|\nabla_x f(x)\|_2^2\ \mu(x)\,dx\Big)$$

We denote $\left(\mathbb{P}(x) - \mathbb{Q}(x)\right)$ as $\mu_1(x)$. To get the optimal $f$, we need to apply KKT conditions on the above equation.

$$L(f, \lambda) = \int_{\mathcal{X}} f(x)\,\mu_1(x)\,dx + \frac{\lambda}{2}\Big(1 - \int_{\mathcal{X}} \|\nabla_x f(x)\|_2^2\ \mu(x)\,dx\Big)$$

From the calculus of variations:

$$L(f + \epsilon h, \lambda) = \int_{\mathcal{X}} (f + \epsilon h)(x)\,\mu_1(x)\,dx + \frac{\lambda}{2}\Big(1 - \int_{\mathcal{X}} \|\nabla_x(f + \epsilon h)(x)\|_2^2\ \mu(x)\,dx\Big)$$

$$= \int_{\mathcal{X}} (f(x) + \epsilon h(x))\,\mu_1(x)\,dx + \frac{\lambda}{2}\Big(1 - \int_{\mathcal{X}} \langle\nabla_x(f + \epsilon h)(x), \nabla_x(f + \epsilon h)(x)\rangle\ \mu(x)\,dx\Big)$$

$$= \int_{\mathcal{X}} (f(x) + \epsilon h(x))\,\mu_1(x)\,dx$$

$$+ \frac{\lambda}{2}\Big(1 - \int_{\mathcal{X}} \big[\|\nabla_x f(x)\|_2^2 + 2\epsilon\langle\nabla_x f(x), \nabla_x h(x)\rangle + \mathcal{O}(\epsilon^2)\big]\ \mu(x) dx\Big)$$

$$= L(f, \lambda) + \epsilon \int_{\mathcal{X}} h(x)\,\mu_1(x)\,dx - \lambda\epsilon \int_{\mathcal{X}} \langle\nabla_x f(x), \nabla_x h(x)\rangle\ \mu(x)\,dx + \mathcal{O}(\epsilon^2)$$

$$= L(f, \lambda) + \epsilon\Big[\int_{\mathcal{X}} h(x)\,\mu_1(x)\,dx - \lambda \int_{\mathcal{X}} \langle\nabla_x f(x), \nabla_x h(x)\rangle\ \mu(x)\,dx\Big] + \mathcal{O}(\epsilon^2)$$

Now we apply integration by part and set $h$ to be zero at boundary as in (Alaoui et al., 2016). We get :

$$\int_{\mathcal{X}} \langle\nabla_x f(x), \nabla_x h(x)\rangle\,\mu(x)\,dx = \int_{\mathcal{X}} \langle\nabla_x f(x)\,\mu(x), \nabla_x h(x)\rangle\,dx$$

$$= \oint_{\partial\mathcal{X}} h(x)\mu(x)\nabla_x f(x).n(x)dS(x)\ - \int_{\mathcal{X}} div\big(\mu(x)\nabla_x f(x)\big)\,h(x)\,dx$$

$$= -\int_{\mathcal{X}} div\big(\mu(x)\nabla_x f(x)\big)\,h(x)\,dx$$

Hence,

$$L(f + \epsilon h, \lambda) = L(f, \lambda) + \epsilon\Big[\int_{\mathcal{X}} \mu_1(x)\,h(x)\,dx + \lambda \int_{\mathcal{X}} div\big(\mu(x)\nabla_x f(x)\big)\,h(x)\,dx\Big] + \mathcal{O}(\epsilon^2)$$

$$= L(f, \lambda) + \epsilon \int_{\mathcal{X}} \Big(\mu_1(x) + \lambda\ div\big(\mu(x)\nabla_x f(x)\big)\Big)\,h(x)\,dx\ + \mathcal{O}(\epsilon^2)$$

The functional derivative of $L(f, \lambda)$, at any test function $h$ vanishing on the boundary:

$$\int_{\mathcal{X}} \frac{\partial L(f, \lambda)}{\partial f}(x)h(x)dx\ =\ \lim_{\epsilon \to 0} \frac{L(f + \epsilon h, \lambda) - L(f, \lambda)}{\epsilon}$$

$$=\ \int_{\mathcal{X}} \Big(\mu_1(x) + \lambda\ div\big(\mu(x)\nabla_x f(x)\big)\Big)\,h(x)\,dx$$

Hence we have:

$$\frac{\partial L(f, \lambda)}{\partial f}(x) = \mu_1(x) + \lambda \ div\big(\mu(x)\nabla_x f(x)\big)$$

For the optimal $f^*, \lambda^*$ first order optimality condition gives us:

$$\mu_1(x) + \lambda^* \ div\big(\mu(x)\nabla_x f^*(x)\big) = 0 \tag{22}$$

and

$$\int_{\mathcal{X}} \|\nabla_x f^*(x)\|^2 \, \mu(x) dx = 1 \tag{23}$$

Note that (See for example (Alaoui et al., 2016)) :

$$div\big(\mu(x)\nabla_x f^*(x)\big) = \mu(x)\Delta_2 f^*(x) + \langle \nabla_x \mu(x), \nabla_x f^*(x)\rangle,$$

since $div(\nabla_x f^*(x)) = \Delta_2 f^*(x)$. Hence from equation (22)

$$\mu_1(x) + \lambda^* \ div\big(\mu(x)\nabla_x f^*(x)\big) = 0$$
$$\Rightarrow \mu_1(x) + \lambda^* \big(\mu(x)\Delta_2 f^*(x) + \langle \nabla_x \mu(x), \nabla_x f^*(x)\rangle\big) = 0$$
$$\Rightarrow \mu_1(x) + \lambda^* \mu(x)\Delta_2 f^*(x) + \lambda^* \langle \nabla_x \mu(x), \nabla_x f^*(x)\rangle = 0$$
$$\Rightarrow \Delta_2 f^*(x) + \left\langle \frac{\nabla_x \mu(x)}{\mu(x)}, \nabla_x f^*(x)\right\rangle + \frac{\mu_1(x)}{\lambda^* \mu(x)} = 0$$
$$\Rightarrow \Delta_2 f^*(x) + \langle \nabla_x \log \mu(x), \nabla_x f^*(x)\rangle + \frac{\mathbb{P}(x) - \mathbb{Q}(x)}{\lambda^* \mu(x)} = 0$$
$$\tag{24}$$

Hence $f^*, \lambda^*$ satisfies :

$$\Delta_2 f^*(x) + \langle \nabla_x \log \mu(x), \nabla_x f^*(x)\rangle + \frac{\mathbb{P}(x) - \mathbb{Q}(x)}{\lambda^* \mu(x)} = 0 \tag{25}$$

and

$$\int_{\mathcal{X}} \|\nabla_x f^*(x)\|^2 \, \mu(x) dx = 1. \tag{26}$$

Let us verify that the optimal critic as found in the geometric definition (Theorem 2) of Sobolev IPM that satisfies:

$$\nabla_i f^*(x) = \frac{\partial f^*(X)}{\partial x_i} = \frac{D^{-i} F_{\mathbb{Q}}(x) - D^{-i} F_{\mathbb{P}}(x)}{\lambda^* d \ \mu(x)} \quad \forall \, i \in [d], \tag{27}$$

satisfies indeed the PDE.

From equation (27), we want to compute $\frac{\partial^2 f(x)}{\partial x_i^2}$ for all $i$:

$$\frac{\partial^2 f(x)}{\partial x_i^2} = \frac{1}{\lambda^* d} \left[ \frac{\mu(x)\big[\frac{\partial}{\partial x_i}(D^{-i} F_{\mathbb{Q}}(x) - D^{-i} F_{\mathbb{P}}(x))\big] - \big[D^{-i} F_{\mathbb{Q}}(x) - D^{-i} F_{\mathbb{P}}(x)\big]\nabla_i \mu(X)}{\mu^2(x)} \right]$$

$$= \frac{1}{\lambda^* d} \left[ \frac{\mu(x)\big[\mathbb{Q}(x) - \mathbb{P}(x)\big] - \big[D^{-i} F_{\mathbb{Q}}(x) - D^{-i} F_{\mathbb{P}}(x)\big]\nabla_i \mu(X)}{\mu^2(x)} \right]$$

$$= \frac{\mathbb{Q}(x) - \mathbb{P}(x)}{\lambda^* d \ \mu(x)} - \frac{\nabla_i \mu(x)}{\mu(x)}\nabla_i f^*(x)$$

Hence,

$$\frac{\partial^2 f(x)}{\partial x_i^2} + \frac{\nabla_i \mu(x)}{\mu(x)}\nabla_i f(x) + \frac{\big(\mathbb{P}(x) - \mathbb{Q}(x)\big)}{\lambda^* d \ \mu(x)} = 0 \tag{28}$$

Adding equation (28) for all $i \in [d]$, we get :

$$\sum_{i=1}^{d} \left( \frac{\partial^2 f(x)}{\partial x_i^2} + \frac{\nabla_i \mu(x)}{\mu(x)}\nabla_i f(x) + \frac{\big(\mathbb{P}(x) - \mathbb{Q}(x)\big)}{\lambda^* d \ \mu(x)} \right) = 0$$

As a result, the solution $f^*$ of the partial differential equation given in equation (25) satisfies the following :

$$\frac{\partial f^*(x)}{\partial x_i} = \frac{D^{-i}F_{\mathbb{Q}}(x) - D^{-i}F_{\mathbb{P}}(x)}{\lambda^* d \ \mu(x)} \quad \forall \, i \in [d]$$

Using the constraint in (26) we can get the value of $\lambda^*$ :

$$\int \|\nabla f^*(x)\|^2 \, \mu(x) \, dx = 1$$

$$\Rightarrow \int \sum_{i=1}^{d} \Big(\frac{\partial f^*(x)}{\partial x_i}\Big)^2 \mu(x) \, dx = 1$$

$$\Rightarrow \lambda^* = \frac{1}{d}\sqrt{\sum_{i=1}^{d} \int \frac{\big(D^{-i}F_{\mathbb{Q}}(x) - D^{-i}F_{\mathbb{P}}(x)\big)^2}{\mu(x)} \, dx} = \mathcal{S}_\mu(\mathbb{P}, \mathbb{Q}).$$

$\square$

*Proof of Theorem 4.* Define the Stein operator (Oates et al., 2017):

$$
\begin{aligned}
T(\mu)[\nabla_x f(x)] &= \frac{1}{2}\langle \nabla_x f(x), \nabla_x \log \mu(x)\rangle + \frac{1}{2}\langle \nabla_x, \nabla_x f(x)\rangle \\
&= \frac{1}{2}\langle \nabla_x f(x), \nabla_x \log \mu(x)\rangle + \frac{1}{2}\Delta_2 f(x).
\end{aligned}
$$

This operator was later used in defining the Stein discrepancy (Gorham & Mackey, 2015; Liu et al., 2016; Chwialkowski et al., 2016; Liu, 2017).

Recall that Barbour generator theory provides us a way of constructing such operators that produce mean zero function under $\mu$. It is easy to verify that:

$$\mathbb{E}_{x \sim \mu} T(\mu)\nabla_x f(x) = 0.$$

Recall that this operator arises from the overdamped Langevin diffusion, defined by the stochastic differential equation:

$$dx_t = \frac{1}{2}\nabla_x \log \mu(x_t) + dW_t$$

where $(W_t)_{t \geq 0}$ is a Wiener process. This is related to plug and play networks for generating samples if the distribution is known, using the stochastic differential equation.

From Theorem 3, it is easy to see that the PDE the Sobolev Critic $(f^*, \lambda^* = \mathcal{S}_\mu(\mathbb{P}, \mathbb{Q}))$ can be written in term of Stein Operator as follows:

$$T(\mu)[\nabla_x f^*](x) = \frac{1}{2\lambda^*}\frac{\mathbb{Q}(x) - \mathbb{P}(x)}{\mu(x)}$$

Taking absolute values and the expectation with respect to $\mathbb{Q}$:

$$|\mathbb{E}_{x \sim \mathbb{Q}}[T(\mu)\nabla_x f^*(x)]| = \frac{1}{2\mathcal{S}_\mu(\mathbb{P}, \mathbb{Q})}\left|\mathbb{E}_{x \sim \mathbb{Q}}\left[\frac{\mathbb{Q}(x) - \mathbb{P}(x)}{\mu(x)}\right]\right|$$

Recall that the definition of Stein Discrepancy :

$$\mathbb{S}(\mathbb{Q}, \mu) = \sup_{\vec{g} \in \mathscr{L}_2(\mathcal{X}, \mu)^{\otimes d}} |\mathbb{E}_{x \sim \mathbb{Q}}[T(\mu)\vec{g}(x)]|$$

It follows that Sobolev IPM critic satisfies:

$$|\mathbb{E}_{x \sim \mathbb{Q}}[T(\mu)\nabla_x f^*(x)]| \leq \mathbb{S}(\mathbb{Q}, \mu),$$

Hence we have the following inequality:

$$\frac{1}{2\mathcal{S}_\mu(\mathbb{P},\mathbb{Q})}\left|\mathbb{E}_{x\sim\mathbb{Q}}\left[\frac{\mathbb{Q}(x)-\mathbb{P}(x)}{\mu(x)}\right]\right|\leq\mathbb{S}(\mathbb{Q},\mu)$$

This is equivalent to:

$$\left|\mathbb{E}_{x\sim\mathbb{Q}}\left[\frac{\mathbb{Q}(x)-\mathbb{P}(x)}{\mu(x)}\right]\right|\leq 2\underbrace{\mathbb{S}(\mathbb{Q},\mu)}_{\text{Stein Good fitness of the model }\mathbb{Q}\text{ w.r.t to }\mu}\underbrace{\mathcal{S}_\mu(\mathbb{P},\mathbb{Q})}_{\text{Sobolev Distance}}$$

Similarly we obtain:

$$\left|\mathbb{E}_{x\sim\mathbb{P}}\left[\frac{\mathbb{Q}(x)-\mathbb{P}(x)}{\mu(x)}\right]\right|\leq 2\underbrace{\mathbb{S}(\mathbb{P},\mu)}_{\text{Stein Good fitness of }\mu\text{ w.r.t to }\mathbb{P}}\underbrace{\mathcal{S}_\mu(\mathbb{P},\mathbb{Q})}_{\text{Sobolev Distance}}$$

For instance consider $\mu=\mathbb{P}$, we have therefore:

$$\frac{1}{2}\left|\mathbb{E}_{x\sim\mathbb{Q}}\left[\frac{\mathbb{Q}(x)}{\mathbb{P}(x)}\right]-1\right|\leq\mathbb{S}(\mathbb{Q},\mathbb{P})\mathcal{S}_\mathbb{P}(\mathbb{P},\mathbb{Q}).$$

*Note that the left hand side of the inequality is not the total variation distance.*

Hence for a sequence $\mathbb{Q}_n$ if the Sobolev distance goes $\mathcal{S}_\mathbb{P}(\mathbb{P},\mathbb{Q}_n)\to 0$, the ratio $r_n(x)=\frac{\mathbb{Q}_n(x)}{\mathbb{P}(x)}$ converges in expectation (w.r.t to $\mathbb{Q}$) to 1. The speed of the convergence is given by the Stein Discrepancy $\mathbb{S}(\mathbb{Q}_n,\mathbb{P})$.

*One important observation here is that convergence of PDF ratio is weaker than the conditional CDF as given by the Sobolev distance and of the good fitness of score function as given by Stein discrepancy.*

$\square$

## C    TEXT EXPERIMENTS: ADDITIONAL PLOTS

**Comparison of annealed versus non annealed smoothing of $\mathbb{P}_r$ in Sobolev GAN.**

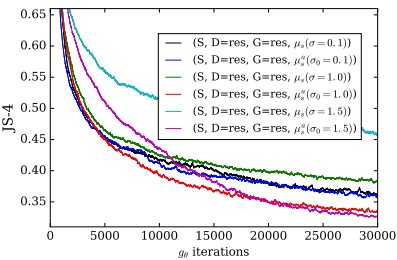

Figure 6: Comparison of annealed versus non annealed smoothing of $\mathbb{P}_r$ in Sobolev GAN. We see that annealed smoothing outperforms the non annealed smoothing experiments.

**Sobolev GAN versus WGAN-GP with RNN.** We fix the generator architecture to Resnets. The experiments of using RNN (GRU) as the critic architecture for WGAN-GP and Sobolev is shown in Figure 7 where we used $\mu=\mu_{GP}$ for both cases. We only apply gradient clipping to stabilize the performance without other tricks. We can observe that using RNN degrades the performance. We think that this is due to an optimization issue and a difficulty in training RNN under the GAN objective without any pre-training or conditioning.

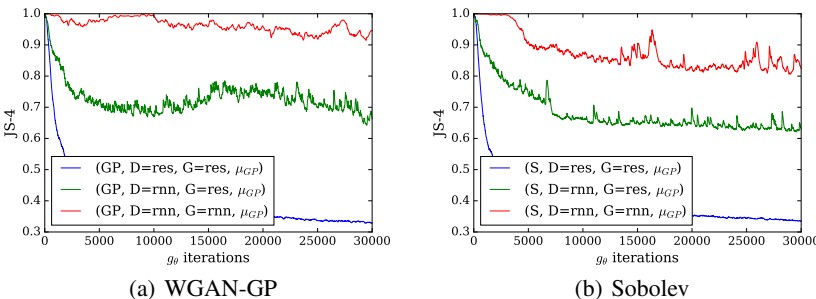

(a) WGAN-GP  (b) Sobolev

Figure 7: Result of WGAN-GP and Sobolev with RNNs.

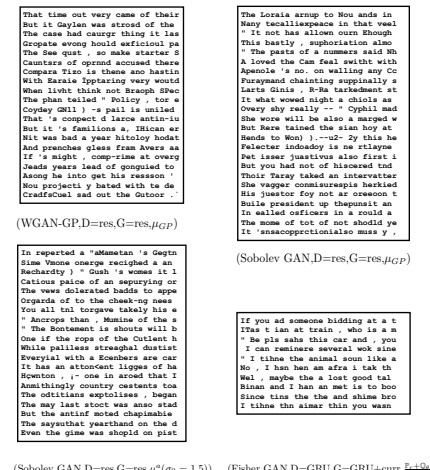

Figure 8: Text samples from various GANs considered in this paper.

## D  SSL: HYPERPARAMETERS AND ARCHITECTURE

For our SSL experiments on CIFAR-10, we use Adam with learning rate $\eta = 2e-4$, $\beta_1 = 0.5$ and $\beta_2 = 0.999$, both for critic $f$ (without BN) and Generator (with BN). We selected $\lambda_{CE} = 1.5$ from $[0.8, 1.5, 3.0, 5.0]$. We train all models for 350 epochs. We used some L2 weight decay: $1e-6$ on $\omega, S$ (i.e. all layers except last) and $1e-3$ weight decay on the last layer $v$. For formulation 1 (Fisher only) we have $\rho_F = 1e-7$, modified critic learning rate $\eta_D = 1e-4$, critic iters $n_c = 2$. For formulation 2 (Sobolev + Fisher) we have $\rho_F = 5e-8$, $\rho_S = 2e-8$, critic iters $n_c = 1$. For the WGAN-GP (Gulrajani et al., 2017) baseline SSL experiment we followed the original paper with critic iters $n_c = 5$, $\eta_G = \eta_D = 1e-4$, Adam $\beta_2$=0.9 and GP weight $\lambda_{GP} = 10.0$. Architectures are as below. We determined $\lambda_{CE} = 0.3$ to be optimal from $[0.03, 0.1, 0.3, 1.0, 3.0]$. As mentioned in Section 6.2, the K+1 critic formulation is not able to fit the training set with the GP constraint, so we fall back to the plain critic formulation where the critic $\langle v, \Phi_\omega(x) \rangle$ does not interact with the classifier $\langle S, \Phi_\omega(x) \rangle$.

Architecture:

```
### CIFAR-10: 32x32. G is dcgan with G_extra_layers=2.
### D is in the flavor of OpenAI Improved GAN, ALI.
G (
  (main): Sequential (
    (0): ConvTranspose2d(100, 256, kernel_size=(4, 4), stride=(1, 1), bias=False)
    (1): BatchNorm2d(256, eps=1e-05, momentum=0.1, affine=True)
    (2): ReLU (inplace)
    (3): ConvTranspose2d(256, 128, kernel_size=(4, 4), stride=(2, 2), padding=(1, 1), bias=False)
    (4): BatchNorm2d(128, eps=1e-05, momentum=0.1, affine=True)
    (5): ReLU (inplace)
```

```
      (6): ConvTranspose2d(128, 64, kernel_size=(4, 4), stride=(2, 2), padding=(1, 1), bias=False)
      (7): BatchNorm2d(64, eps=1e-05, momentum=0.1, affine=True)
      (8): ReLU (inplace)
      (9): Conv2d(64, 64, kernel_size=(3, 3), stride=(1, 1), padding=(1, 1), bias=False)
      (10): BatchNorm2d(64, eps=1e-05, momentum=0.1, affine=True)
      (11): ReLU (inplace)
      (12): Conv2d(64, 64, kernel_size=(3, 3), stride=(1, 1), padding=(1, 1), bias=False)
      (13): BatchNorm2d(64, eps=1e-05, momentum=0.1, affine=True)
      (14): ReLU (inplace)
      (15): ConvTranspose2d(64, 3, kernel_size=(4, 4), stride=(2, 2), padding=(1, 1), bias=False)
      (16): Tanh ()
   )
)
D (
  (main): Sequential (
    (0): Dropout (p = 0.2)
    (1): Conv2d(3, 96, kernel_size=(3, 3), stride=(1, 1), padding=(1, 1))
    (2): LeakyReLU (0.2, inplace)
    (3): Conv2d(96, 96, kernel_size=(3, 3), stride=(1, 1), padding=(1, 1), bias=False)
    (5): LeakyReLU (0.2, inplace)
    (6): Conv2d(96, 96, kernel_size=(3, 3), stride=(2, 2), padding=(1, 1), bias=False)
    (8): LeakyReLU (0.2, inplace)
    (9): Dropout (p = 0.5)
    (10): Conv2d(96, 192, kernel_size=(3, 3), stride=(1, 1), padding=(1, 1), bias=False)
    (12): LeakyReLU (0.2, inplace)
    (13): Conv2d(192, 192, kernel_size=(3, 3), stride=(1, 1), padding=(1, 1), bias=False)
    (15): LeakyReLU (0.2, inplace)
    (16): Conv2d(192, 192, kernel_size=(3, 3), stride=(2, 2), padding=(1, 1), bias=False)
    (18): LeakyReLU (0.2, inplace)
    (19): Dropout (p = 0.5)
    (20): Conv2d(192, 384, kernel_size=(3, 3), stride=(1, 1), bias=False)
    (22): LeakyReLU (0.2, inplace)
    (23): Dropout (p = 0.5)
    (24): Conv2d(384, 384, kernel_size=(3, 3), stride=(1, 1), bias=False)
    (26): LeakyReLU (0.2, inplace)
    (27): Dropout (p = 0.5)
    (28): Conv2d(384, 384, kernel_size=(1, 1), stride=(1, 1), bias=False)
    (30): LeakyReLU (0.2, inplace)
    (31): Dropout (p = 0.5)
  )
  (V): Linear (6144 -> 1)
  (S): Linear (6144 -> 10)
)
```

