# OpenReview forum: "Sobolev GAN"
_ICLR.cc/2018/Conference — Accept (Poster)_

### Official Review · AnonReviewer2 · 2017-11-27
**The paper suggests a novel regularization scheme for GANs based on a Sobolev norm, measuring deviations between L2 norms of derivatives. It establishes some nice theoretical properties, and demonstrates effectiveness through simulations.**

**Rating:** 8
**Confidence:** 4

**Review:**

The paper deals with the increasingly popular GAN approach to constructing generative models.  Following the first formulation of GANs in 2014, it was soon realized that the training dynamics was highly unstable, leading to significant difficulties in achieving stable results. The paper by Arjovsky et al (2017) provided a framework based on the Wasserstein distance, a distance measure between probability distributions belonging to the class of so-called Integral Probability Metrics (IPMs). This approach solved the stability issues of GANs and demonstrated improved empirical results. Several other works were then developed to deal with these stability issues, specifically the Fisher IPM. Both these methods relied on discriminating between distributions P and Q based on computing a function f, belonging to an appropriate function class {\cal F}, that maximizes the deviation E_{x~P}f(x)-E_{x~Q}f(x). The main issue relates to the choice of the class {\cal F}. For the Wasserstein distance this was the class of L_1 Lipschitz functions, while for the Fisher distance it was the class of square integrable functions. The present paper introduces a new notion of distance, where {\cal F} is the defined through the Sobolev norm, based on the L_2 norm of the gradient of f(x), with respect to a measure \mu(x), where the latter can be freely chosen under certain assumptions.

The authors prove a theorem related to the properties of the Sobolev norm, and express it in terms of the component-wise conditional distributions. Moreover, they show that the optimal critic f is obtained by solving a PDE subject to zero boundary conditions. They then use their suggested metric in order to develop a GAN algorithm, and present experimental results demonstrating its utility. The Sobolev IPM has two nice features. First, it is based on the component-wise conditional distribution of the CDFs, and, second, its relation to the Laplacian regularizer from manifold learning. Its 1D version also relates to the well-known von Mises Cramer statistics used in hypothesis testing.

The paper belongs to a class of recent papers attempting to suggest improvements to the original GAN algorithm, relying on the KL divergence. It is well conceived and articulated, and provides an interesting and potentially powerful new direction to improve GANs in practice. However, it is somewhat difficult to follow the paper, and would urge the authors to improve and augment their presentation of the following issues.
1)	One often poses regularization schemes based on optimality criteria. Is there any optimality principle under which the Sobolev IPM is a desired choice?
2)	The authors argue that their approach is especially well suited for discrete sequential data. This issue was not clear to me, and it would be good if the authors could expand on this issue and provide a clearer explanation.
3)	How would the Sobolev norm behave under a change of coordinates or a homeomorphism of the space? Would it make sense to require some invariance in this respect?
4)	The Lagrangian in eq. (9) contains both a Lagrange constraint on the Sobolev norm and a penalty term. Why are both needed? Why do the updates of  \lambda and p in Algorithm 1 used different schemes (SGD and ADAM, respectively).
5)	Table 2, p. 13 – it would be nice to see a comparison to the recently introduced gradient penalty approach, Gulrajani et al., Improved training of wasserstein gans. arXiv preprint arXiv:1704.00028, 2017.
6)	The integral defining F_p(x) on p. 3 has x as an argument on the LHS and as an integrand of the RHS. Please correct this. Also specify that x=(x_1,\ldots,x_d).

---

> ### Author Response · Authors · 2017-12-20
> **WGAN-GP SSL added to the paper/ Implicit conditioning for sequential data models the context**
>
> We thank the reviewer for his encouraging and thoughtful comments and his suggestions that we already incorporated in the revision. We address below his main questions:
>
> 1- It would be interesting to find a primal formulation that has sobolev IPM as a dual. Given the PDE that the critic of Sobolev satisfies, one way to find in which optimal way the Sobolev discrepancy is defined, is to attempt to write its
> dynamic form. While we don’t have yet a formal proof (we are working on it) we conjecture that sobolev IPM may relate to the following dynamic problem:
>
>  inf_{f_t in W_2,0(q_t)} \int _{0}^T \nor{\nabla_x f^t(x)} q_t(x) dx
>               d(q_t) /dt = -div(q_t \nabla_x f^t(x))
>              q_0=Q , q_T=P
> Note that Wasserstein 2 distance has a similar form due to Benamou et al ("A computational fluid mechanics solution to the Monge-Kantorovich mass transfer problem" by Jean-David Benamou et al.) and has instead of \nabla_x f , a function g \in \mathbb{R}^d in L_2(q_t). We think Sobolev IPM looks for “smooth flows” to move density masses.  The effort for moving the mass is measured with the sobolev norm of critics and we wish to minimize this effort.
>
> 2- When we wish to compare discrete sequences we face two problems:
>
> a- The generator is continuous and the real data is discrete. Wasserstein distance is well known to address this issue of discrete/continuous matching. Divergences like KL or JS compare density functions and hence are not suited for this task.  For sobolev IPM: by noising the discrete data (smoothing)  we match two continuous distributions based on the coordinatewise conditional CDFs. Since the noising is annealed over training we will end up matching the discrete and continuous distributions. CDFs are better suited for the discrete nature of the problem as shown in Figure 1 in the revision of the paper.
>
> b - another problem is that we need for sequences a discrepancy between processes, rather than just distributions. For time processes, how the joint distributions factors in terms of conditioning is very important. Sobolev IPM gives a little advantage by comparing for each coordinate a unique quantity CDF(x_i| x^{-i})PDF(x^{-i}), since the Sobolev discrepancy captures some conditioning it has the ability of measuring coordinate dependencies, crucial for sequence/time process modeling.
>
> See also our answer to Reviewer 1, part 3. We included Figure 1 in the paper revision to illustrate this.
>
> 3- Behavior under coordinate change:
>
> Discrepancy is unchanged but critic rotated: Assume we rotate P and Q and use mu=P+Q/2, the discrepancy won’t change but the critic gradients will be rotated.
>
> Incorporating invariance in mu: Since mu is free, we think a coordinate change might be a good place for  incorporating desired invariances. for instance if we set d\mu(gx) = \frac{P( gx) + Q(g x) }{2} dg dx, g is an element in a lie group,  our critic will be smooth and well defined on the support of  an “augmented” distribution, which will probably benefit the semi-supervised learning with sobolev GAN. This is in the spirit of the recent work on invariances with GAN from Kumar et al NIPS 2017.
>
> Kumar et al  NIPS 2017. Semi supervised learning with GANs: Manifold invariance with improved inference.
>
> 4- Augmented Lagrangian: We are using an augmented Lagrangian to enforce the equality constraint. Hence we have a lagrangian and a penalty  term. Note that in Benamou paper an augmented lagrangian was also used. We used adam for optimizing the critic and the generator which is default practice, while the simple SGD for the lagrangian is following common practice in ADMM (convex optimization) where also our penalty coefficient is the learning rate for the lagrangian (see eg slide 9 of https://web.stanford.edu/~boyd/papers/pdf/admm_slides.pdf)
>
> 5- The WGAN-GP paper did not provide SSL results; in our experiments we did not get WGAN-GP to work for SSL (same experimental setup). We now report those results in Table 2 page 15.
>
> 6- This was fixed in the revision thanks for pointing it out.

---

### Official Review · AnonReviewer1 · 2017-12-01
**Interesting mathematically, unclear if a step above other GANs**

**Rating:** 6
**Confidence:** 3

**Review:**

Summary: The authors provide another type of GAN--the Sobolev GAN--which is the typical setup of a GAN but using a function class F for which f belongs to F iff \grad f belongs to L^2(mu). They relate this MMD to the Cramer and Fisher distance and then produce a recipe for training GANs with this sort of function class. In their empirical examples, they show it has similar performance to the WGAN-GP.

Overall, the paper has some interesting mathematical relationships to other MMDs. However, I finished reading the paper wondering why one would want to trust this GAN over any of the other GANs. I may have missed it, but I didn't see any compelling theoretical reason the gradients from this method would prove superior to many of the other GANs in existence today. The authors argue "from [equation 5] we see that we are comparing CDFs, which are better behaved on discrete distributions," but I wasn't sure what exactly to make of this comment.

Nits:
* The "Stein metric" is actually called the Stein discrepancy [see Gorham & Mackey (2015) Measuring Sample Quality using Stein's Method].

---

> ### Author Response · Authors · 2017-12-20
> **Empirical performance (improved SSL) and compelling theoretical reason of the gradients of Sobolev GAN (fokker planck and sobolev approximation)**
>
> We thank the reviewer for his comments and address their main concerns: (1) empirical performance and (2) compelling theoretical reasons for superior gradients. We uploaded a revision of the paper to incorporate the reviewer comments.
>
> - Empirical  performance similar to WGAN-GP: agreed for text modeling, but not for SSL. (1) In text generation, our goal was to understand the success of WGAN-GP. The sobolev point of view gives us the insight that the gradient penalty is not only introducing stability to the training, but also providing implicit conditioning needed for sequence modeling. (2) In SSL, while WGAN-GP fails, we show how to use sobolev IPM in SSL and link it to laplacian regularization in manifold learning (Please see updated Table 2 page 15 where we added a comparison to WGAN-GP)
>
> - Gradients of sobolev critic :
>
> 1- Convergence of critics in Sobolev Sense:  One interesting property we show (Lemma 1 originally in appendix, we moved it now to the main paper) is about the approximation of the optimal critic f^*,  when f is parameterized in a hypothesis class (i.e. neural network, f_H).
> Most GANs don’t have any guarantee.
> - Fisher GAN provides an approximation  guarantee in Lebesgue sense, meaning norm(f^* - f_H)..
> - Sobolev GAN provides an approximation guarantee in Sobolev sense, meaning norm (grad f^* - grad f_H).
> Having in mind that the gradient of the critic is the information that is passed on to the generator, we see that this convergence in Sobolev sense to the optimal critic is an important property for GAN training.  We  highlighted this in the revision of the paper .
>
> 2- Meaningful Gradient Directions by fokker planck diffusion: We show that the optimal  critic satisfies a PDE that relates to a deterministic fokker planck diffusion (https://en.wikipedia.org/wiki/Fokker–Planck_equation). Assume our goal is to move a density Q to a density P.  we start by computing the sobolev critic between Q and P. If we have particles X_t whose initial distribution distributions Q_0= Q and we move those particles with the gradient of the critic (X_t=X_{t-1}+ epsilon \nabla_x f^t_{q_t,P}(x) ).  we do this process for T steps by  recomputing the critic at each time between the particle distributions q_t and P.  The relation of Sobolev IPM to fokker planck diffusion gives us the evolution of the density of the moving particles. At the end of the process we are guaranteed by fokker planck diffusion to converge to the distribution P. Note that this diffusion has two steps: compute the critic between q_t and P, and update the particles X_t. This is similar to the gradient descent applied to learning GANS: compute critic, and update particles, with the main difference that we are working here with densities, in GAN the generator distribution is degenerate and does not have a density.
>
> For an illustration of how we transport Q to P using critic gradients, we provide these examples:
> one is using sobolev IPM critic that has the diffusion property (https://www.dropbox.com/s/ymwmddvsdpj29lq/sobolev_near_mu_q_full_65.mp4?dl=0 ) and one with the MMD distance critic (https://www.dropbox.com/s/ehpmhfghh5wbo5n/mmd_near_65.mp4?dl=0) that does not have this diffusion property. In those videos we see how the pink particles (Q) move to have same density as the black particles (P). In each frame the particles move with the gradient of the critic. The critic is recomputed between 2 frames, between the new density q_t and P in black. The level sets of the new critic are shown in each frame.
> After 65 iterations with the same learning rate, we see that sobolev IPM succeeds at this task, while the MMD fails. One can see that the descent on sobolev discrepancy converges almost completely in just 65 iterations but the same is not true for MMD based descent method. Although, MMD based descent method takes almost 300 iterations to converge which can be seen in the video link given below : https://www.dropbox.com/s/shffwn3tffvb52r/mmd_near_full.mp4?dl=0

---

> > ### Author Response · Authors · 2017-12-20
> > **implicit conditioning and CDF for smoothed discrete /continuous matching**
> >
> > 3- CDF for continuous/discrete matching: Note that when we want to generate text, we are facing a problem of a continuous generator and discrete real data. If we were to compare a continuous and a discrete distribution based on pdfs, this  will fail (e.g.  KL, JSD, etc). Wasserstein distance for instance in one dimension  is known to be the comparison of inverse CDFs  and this makes it possible to compare continuous and discrete distributions.  For sobolev IPM: by noising the discrete data we match two continuous distributions based on the coordinatewise conditional CDFs. Since the noising is annealed over training we will end up matching the discrete and continuous distributions. We hypothesise that the implicit conditioning (coordinatewise conditional CDF matching) implied by the gradient regularizer allows end to end training of GAN text generation, since we are implicitly conditioning on the context for each coordinate. We added plots to illustrate this (Figure 1 in the revision).

---

### Official Review · AnonReviewer4 · 2017-12-04
**Not empirically better than WGAN-GP**

**Rating:** 6
**Confidence:** 4

**Review:**

The paper proposes a different gradient penalty for GAN critics.
The proposed penalty is forcing the expected squared norm of the gradient to be equal to 1.
The corresponding integral probability metric is well analysed.

Pros:
- The paper provides a nice overview of WGAN-GP, Fisher GAN and Sobolev GAN.
The differences and similarities and mentioned.
- The paper shows that Sobolev IPM is comparing coordinate-wise conditional CDFs.
- The 1D example in Section 4.2 shows a limitation of Fisher GAN.

Cons:
- The introduced gradient penalty is harder to optimize.
Algorithm 1 is using a biased estimate of the penalty.
If having independent samples in the minibatch,
it would be possible to construct an unbiased estimate of the penalty.
- An unbiased estimate of the gradient penalty
will be hard to construct when not having two independent real samples.
E.g., when doing conditional modeling with a RNN.
- The algorithm requires to train the critic well
before using the critic.
The paper does not provide an improvement over WGAN-GP in this direction.
MMD GAN and Cramer GAN may require less critic training steps.
- The experimental results do not demonstrate an improvement over WGAN-GP.
- Too much credit is given to implicit conditioning.
The Jensen Shanon divergence can be also written as a chain
of coordinate-wise JS divergences. That does not guarantee non-zero gradients
from the critic. A critic with non-zero gradients seems to be more important.


Minor typos:
s/pernalty/penalty/
s/ccordinate/coordinate/

---

> ### Author Response · Authors · 2017-12-20
> **empirical comparaison to WGAN-GP : on par for text, Sobolev is better in SSL**
>
> We thank the reviewer for his  comments.  We have uploaded a revision of the paper to incorporate the reviewer comments.
>
> First we stress that the goal of this paper is to better *understand* the gradient penalty and to know what it adds to the learning problem,  rather than an emphasis on performance, although we show better performance than WGAN-GP in semi-supervised learning (we will expand on this later).
> WGAN-GP is the  first paper to show text GAN working end to end, our goal was to explain what is behind this success.
>
> We hope to answer the main concerns of the reviewer:
>
> 1) Biased estimate of the penalty. Reviewer 4 thinks reusing the same samples between objective and constraint introduces bias, however the use of the same samples in the loss and the constraint is a well-studied problem and is not an issue theoretically and practically. See for example (Shivaswamy and Jebara JMLR 2010  http://www.jmlr.org/papers/volume11/shivaswamy10a/shivaswamy10a.pdf).
> The Lagrangian formulation  does not  need another minibatch for an unbiased estimate, while the quadratic penalty theoretically needs one. But using some concentration inequalities for data dependent constraints as done in (Shivaswamy and Jebara, Section 4.6), one can show that this bias is very small and does not add any difficulty to the optimization. In Shivaswamy and Jebara,  the term “landmarks” is used for the samples in the constraint. It is shown in this work that using the same samples for the loss and the constraints, introduces a small bias that vanishes with the number of samples. We will add a discussion in the paper.
> Furthermore we did not find empirically any difficulty in optimizing Sobolev GAN. (we added remark 1 under Algorithm 1 to refer to the small bias and to Shivaswamy and Jebara )
>
>
> 2) From a theoretical perspective, all GAN formulations (including MMD GAN and Cramer GAN) require full maximization of the discriminator (critic) in the inner loop (Arjovsky and Bottou ICLR 2017). In practice, usually a small number of iterations n_c is used instead in the inner loop (disc maximization).
> As stated in Appendix D, empirically we find that we can use n_c= 1 or 2 in Sobolev GAN.
>
> 3) Performance: WGAN-GP has not shown any semi-supervised learning performance. In our experiments, WGAN-GP for semi supervised learning gives bad performance (see updated Table 2 page 15). We show in this paper how to make use of the sobolev norm to regularize SSL in IPM based SSL, and we show the connection to laplacian regularization in manifold learning.
>
> 4A) Example in one D between 2 Diracs is not only a limitation of Fisher IPM , it is a limitation of any distance comparing PDFs as shown in Arjovsky et al. This is in line with the intuition of the reviewer on the importance of non zero gradients.
>
> 4B) Implicit conditioning: the Reviewer may have missed the point of the implicit conditioning introduced by Sobolev IPM (coordinate-wise conditional CDF form, equation 5). Indeed the Bayes rule can be used to rewrite JS coordinate wise, nevertheless this conditioning  acts on PDF (probability density functions): it compares PDF{x_i|x^{-i}} PDF(x^{-i}), which is  equal for all coordinate to PDF(x_1,dots x_d). Note that the Sobolev IPM compares instead  *unique quantities for each coordinate* CDF{x_i|x^{-i}} PDF(x^{-i}) (see Equation 5), making the model able to learn coordinate conditional distributions, in other words the ability to model context. The empirical evidence for the benefit of implicit conditioning is in Section 6.1 where we show when training Fisher GAN for text generation we need curriculum conditioning, while Sobolev GAN doesn’t. We highlight there the crucial role of conditioning in sequence learning, it is not only a vanishing gradient problem. The gradient penalty in Sobolev GAN and WGAN-GP imply this implicit conditioning that is crucial in end to end sequence/ text generation using GAN.

---

> > ### Public Comment · ~Leon_Boellmann1 · 2017-12-20
> > **GANs do not require maximization of the discriminator in the inner loop**
> >
> >  Just pass by and want to comment about your bullet point (2). In the original paper by Ian Goodfellow, they prove the theoretical convergence if the discriminator is maximized in the inner loop. Actually this is not needed. From the theoretical perspective, simultaneous gradient descent can guarantee convergence (under some conditions). Details can be found in the following two papers:
> >
> > 1. Gradient descent GAN optimization is locally stable, NIPS 2017.
> >
> > 2.  GANs Trained by a Two Time-Scale Update Rule Converge to a Local Nash Equilibrium, NIPS 2017.

---

> > > ### Author Response · Authors · 2017-12-20
> > > **clarifications**
> > >
> > > There are three meanings of convergence in the GAN context:
> > > a- convergence of the min-max game to an equilibrium point in the parameter space of the generator and the critic.
> > > b- convergence of f_omega to sup_f in the inner loop - necessary to achieve equivalence to closed form probability metric.
> > > c- convergence of fake (generator) to real distribution
> > >
> > > We are referring to type (b) convergence. What you are referring to as "stability" or "convergence" is type (a) convergence of the game to an equilibrium point  (convergence of w and theta) to an equilibrium. Local convergence to an equilibrium is important for stability, otherwise we can have some oscillation and have no convergence of gradient descent to a stable saddle point of the cost function as explored in this paper https://arxiv.org/pdf/1711.00141.pdf.
> > >
> > > Unfortunately there is no formal guarantee  that type (a)  convergence implies type (c) convergence. Original Goodfellow paper, Arjovsky and bottou ICLR 2017, were concerned in type (c) convergence. Under some conditions of non vanishing gradient and some assumption on the generator one can show that type (b) convergence, ensure that we are minimizing the probability metric at hand . We think it is still an open question theoretically the interplay between types (a), (b) and (c) convergence and what are the implication on the inner loop optimization of the discriminator and on  the architectures of the generator and discriminator.

---

> > ### Comment · AnonReviewer4 · 2017-12-27
> > **Too much credit for implicit conditioning**
> >
> > Thank you for your response and clarifications.
> > The analysis of Sobolev IPM would deserve to be published.
> > I increased my rating to: "6: Marginally above acceptance threshold".
> >
> > The paper would still benefit from rewording a few sentences.
> > For example, I do not find the following sentences helpful:
> > "Matching conditional dependencies between coordinates is crucial for sequence modeling."
> > "We validate that the conditioning implied by Sobolev GAN is crucial for the success and stability of GAN in text generation."
> >
> > Your illustrative example from Section 4.2 nicely shows that Fisher IPM does not provide a useful training signal
> > when measuring the distance between two distributions with disjoint supports.
> > I would not call this problem a lack of "implicit conditioning".
> > The Wasserstein GAN paper compared different divergences by looking at convergence of sequences of probability distributions. That seems to be a more general approach than requiring an implicit conditioning.
> > It is not clear what form of implicit conditioning is needed and which implicit conditioning is done by the Wasserstein distance.

---

> > > ### Author Response · Authors · 2018-01-05
> > > **clarification on implicit conditioning: curriculum conditioning for fisher versus implicit conditioning in  Sobolev (Figure 4 in the paper)**
> > >
> > > We thank the reviewer for their comment. We will consider rephrasing those sentences for clarity. The success of Fisher GAN with curriculum conditioning on text generation supports that conditioning is the missing ingredient (as we show in Figure 4 in the paper). The explicit curriculum conditioning parallels the implicit conditioning induced by the Sobolev IPM, where context modeling is induced by the metric.
> > > Analysis using sequences of probability distributions, and investigating similar implicit conditioning for WGAN-GP would be interesting directions for future work.

---

> > > > ### Comment · AnonReviewer4 · 2018-01-05
> > > > **Different conclusion**
> > > >
> > > > Your experiments with curriculum conditioning can be interpreted also differently.
> > > > 1) The recurrent discriminator was helping only slightly.
> > > > So an explicit conditioning in the loss computation is not sufficient.
> > > >
> > > > 2) The curriculum for the generator may be helping for a different reason.
> > > > If the discriminator is not able to perfectly recognize real and generated examples,
> > > > the generator gets a non-zero training signal, even if using Fisher GAN.

---

### Official Review · AnonReviewer3 · 2017-12-05
**Good effort towards a better GAN**

**Rating:** 7
**Confidence:** 3

**Review:**

This paper designs a new IPM(Integral Probability Metric) that uses the gradient properties of the test function. The advantage of Sobolev IPM over the Fisher IPM is illustrated by the insight given in Section 4.2. This is convincing. For comparing the true distribution and the generated distribution, it is much better to provide a quantitative measurement, rather than a 0-1 dichotomy. This target is implicitly achieved by the reproducing kernel methods and the original phi-divergence.

On the other side, the paper is hard to follow, and it contains many long sentences, some 4-5 lines. The formulation of the Sobolev norm could be improved at the top of page 6.

---

> ### Author Response · Authors · 2017-12-20
> **we improved the presentation**
>
> We thank the reviewer for his encouraging and supportive comments. We have revised the paper and we improved the presentation of the sobolev IPM on top of page 6 and added additional theoretical results in this section, as well as more illustrative examples (Figure 1).

---

### Public Comment · ~Leon_Boellmann1 · 2017-12-05
**Comparison with other GANs**

 Dear authors,
 After I read this paper, I have a couple questions:
 1. There are so many different divergent metrics for distributions, and each of them corresponds to a specific GAN. What is the advantage of one over another? Is there one that has a dominating performance?
 2. Instead of evaluating each variant of GAN one by one (we already  have 7 variants as on the list, and I can foresee more by changing the divergence metrics, say Wasserstein-p distance can result in a new GAN ), do we have a general framework that incorporates all these variants of GANs? For example, the framework has some parameters that we can tune to change the form of GANs. It is very useful in practice, because these parameters together with the network parameters can serve as hyperparameters in the performance tuning.
3. It points out in Section 4.2 that the CDF comparison is more suitable than PDF for comparing distributions on discrete spaces, because W(P,Q) = |a1-a2| and F(P,Q) = 2. May I know the logic why F(P,Q) = 2 is better than W(P,Q) = |a1-a2|? Is there a toy example that shows Wasserstein GAN does not work, while Sobolev GAN works?

Thanks!

---

> ### Author Response · Authors · 2017-12-20
> **comparaison to other GANs**
>
> Thank you for your interest and your questions!
>
> 1 and 2 - Unfortunately there is no easy answer to your question “which GAN is better”. It will be an empirical question given the specific application at hand. For example it has been observed that sample quality and semi-supervised performance may negatively impact each other (Bad GAN: Dai et al. NIPS 2017).
>
> 1- There are two big families of discrepancies: f-divergences and IPMs (Integral probability metrics). They are all valid for GAN training. For using them in GAN training how the critic is regularized impacts both the metric being computed and the stability of GAN training. For instance weight clipping performances poorly, while gradient penalty performs quite well. Variance control as in fisher gan performs well also in the continuous case. Spectral normalization introduced recently seems also to perform well. Our understanding until now is the main advantage of gradient penalty is 1) numerical/stability: a better control of the gradient of the critic that is passed on by backpropagation to the generator.  2) theoretical : introduces some factoring/ implicit conditioning crucial for sequence (for e.g text) modeling.
>
> 2- It is possible to combine regularizers under the IPM objective, which we showed in this paper (variances and gradients i.e fisher and sobolev) to achieve good performance in SSL , without the need of any batch or layer normalization. Combining losses is a plausible future direction.
>
> 3.  We are comparing fisher to sobolev in the discrete case, the intuition being that when we have disjoint supports Fisher may not provide good gradients for the generator. We are not claiming that Wasserstein would not work, rather we are saying that to achieve the exact Wasserstein distance, the Lipschitz  constraint is not easy to enforce, while for Sobolev IPM the constraint is computationally tractable.

---

> > ### Public Comment · ~Leon_Boellmann1 · 2017-12-20
> > **Thanks!**
> >
> > Thanks for your reply!

---

### Public Comment · ~gonzalo_e_mena1 · 2017-12-13
**Question on Table 1**

Thanks to the authors for the paper, I really appreciate this attempt to unify ideas that have been around lastly (at least this is my interpretation after a  not-in-depth look at the paper).
My question is about Table 1: It is a great table, but I wonder why there is a NA in the closed form expression for the Wasserstein IPM: why dont use what is known from Kantorovich duality? (i.e inf_{Z=(X,Y) in U(P,Q) E_Z(|X-Y|)) where U(P,Q) is the set of joints Z that are consistent with the marginals P,Q ?)

Thanks

---

> ### Author Response · Authors · 2017-12-20
> **Table 1**
>
> Thanks for your interest and your comment! Indeed we tried to summarize in TABLE 1 the recent discrepancies used in the GAN literature to put in context our contribution on Sobolev IPM. We did not put the primal formulation of wasserstein under closed form, meaning a formula that can be readily computed given the densities of the two distributions: The primal  still needs to be solved using entropic regularization or a  linear program (LP) to find the (regularized) wasserstein distance. We can probably add the primal and state that the regularized wasserstein can be solved using entropic regularization with the sinkhorn algorithm (Cuturi et al. 2013), there was some recent GANs using the primal formulation and automatic differentiation through the sinkhorn algorithm, we will add those to the paper. We updated Table 1 to reflect GANs using the primal formulation of optimal transport.

---

### Author Response · Authors · 2017-12-20
**General comment : We uploaded a minor revision of the paper to incorporate reviewers comments**

The main changes are:
- added WGAN GP performance on SSL (Table 2 page 15)
- moved Lemma 1 from appendix to the main paper for the approximation of the sobolev IPM in a hypothesis class (page 7) for showing that the approximation error is in the sobolev sense.
- added Figure 1 to illustrate CDF versus PDF based matching ( Figure 1 page 8)

---

### Decision · Program_Chairs · 2018-01-29
**ICLR 2018 Conference Acceptance Decision**

**Decision:**

Accept (Poster)

**Comment:**

The paper provides a useful analysis of the role of gradient penalties and the performance of the proposed approach in semi-supervised cases.